# Revisiting Contrastive Methods for Unsupervised Learning of Visual Representations

**Wouter Van Gansbeke**[*,1]   **Simon Vandenhende**[*,1]   **Stamatios Georgoulis**[2]   **Luc Van Gool**[1,2]

[1] KU Leuven/ESAT-PSI    [2] ETH Zurich/CVL

## Abstract

Contrastive self-supervised learning has outperformed supervised pretraining on many downstream tasks like segmentation and object detection. However, current methods are still primarily applied to curated datasets like ImageNet. In this paper, we first study how biases in the dataset affect existing methods. Our results show that an approach like MoCo [19] works surprisingly well across: (i) object- versus scene-centric, (ii) uniform versus long-tailed and (iii) general versus domain-specific datasets. Second, given the generality of the approach, we try to realize further gains with minor modifications. We show that learning additional invariances - through the use of multi-scale cropping, stronger augmentations and nearest neighbors - improves the representations. Finally, we observe that MoCo learns spatially structured representations when trained with a multi-crop strategy. The representations can be used for semantic segment retrieval and video instance segmentation without finetuning. Moreover, the results are on par with specialized models. We hope this work will serve as a useful study for other researchers. The code and models are available [2].

## 1   Introduction

Self-supervised learning (SSL) [25] aims to learn powerful representations without relying on human annotations. The representations can be used for various purposes, including transfer learning [19], clustering [1, 43, 44] or semi-supervised learning [6]. Recent self-supervised methods [4, 5, 17, 19] learn visual representations by imposing invariances to various data transformations. A popular way of formulating this idea is through the instance discrimination task [47] - which treats each image as a separate class. Augmentations of the same image are considered as positive examples of the class, while other images serve as negatives. To handle the large number of instance classes, the task is expressed as a non-parametric classification problem using the contrastive loss [18, 34].

Despite the recent progress, most methods still train on images from ImageNet [12]. This dataset has specific properties: (1) the images depict a single object in the center of the image, (2) the classes follow a uniform distribution and (3) the images have discriminative visual features. To deploy self-supervised learning into the wild, we need to quantify the dependence on these properties. Therefore, in this paper, we first study the influence of dataset *biases* on the representations. We take a utilitarian view and transfer different representations to a variety of downstream tasks.

Our results indicate that an approach like MoCo [19] works well for both object- and scene-centric datasets. We delve deeper to understand these results. A key component is the augmentation strategy which involves random cropping. For an object-centric dataset like ImageNet, two crops from the same image will show a portion of the same object and no other objects. However, when multiple

---

[*]Equal contribution

[2]Code: https://github.com/wvangansbeke/Revisiting-Contrastive-SSL

objects are present, a positive pair of non-overlapping crops could lead us to wrongfully match the feature representations of different objects. This line of thought led recent studies [38, 39] to believe that contrastive SSL benefits from object-centric data.

So how can contrastive methods learn useful representations when applied to more complex, scene-centric images? We propose a hypothesis that is two-fold. First, the default parameterization of the augmentation strategy avoids non-overlapping views. As a result, positive pairs will share information, which means that we can match their representations. Second, when applying more aggressive cropping, we only observe a small drop in the transfer learning performance. Since patches within the same image are strongly correlated, maximizing the agreement between non-overlapping views still provides a useful learning signal. We conclude that, in visual pretraining, combining the instance discrimination task with random cropping is universally applicable.

The common theme of recent advances is to learn representations that are invariant to different transformations. Starting from this principle, we try to improve the results obtained with an existing framework [19]. More specifically, we investigate three ways of generating a more diverse set of positive pairs. First, the multi-crop transform from [4] is revisited. Second, we examine the use of a stronger augmentation policy. Third, we leverage nearest neighbors mined online during training as positive views. The latter imposes invariances which are difficult to learn using handcrafted image transformations. The proposed implementation requires only a few lines of code and provides a simple, yet effective alternative to clustering based methods [4, 30]. Each of the proposed additions is found to boost the performance of the representations under the transfer learning setup.

The multi-crop transform realizes significant gains. We probe into what the network learns to explain the improvements. The multi-crop transform maximizes the agreement between smaller (local) crops and a larger (global) view of the image. This forces the model to learn a more spatially structured representation of the scene. As a result, the representations can be directly used to solve several dense prediction tasks without any finetuning. In particular, we observe that the representations already model class semantics and dense correspondences. Furthermore, the representations are competitive with specialized methods [24, 53]. In conclusion, the multi-crop setup provides a viable alternative to learn dense representations without relying on video data [24, 29] or handcrafted priors [23, 44, 53].

In summary, the overall goal of this paper is to learn more effective representations through contrastive self-supervised learning without relying too much on specific dataset biases. The remainder of this paper is structured as follows. Section 2 introduces the framework. Section 3 whether the standard SimCLR augmentations transfer across different datasets. This question is answered positively. Based upon this result, Section 4 then studies the use of additional invariances to further improve the learned representations. We hope this paper will provide useful insights to other researchers. .

## 2  Framework

We briefly introduce the contrastive learning framework. The idea is to generate feature representations that maximize the agreement between similar (*positive*) images and minimize the agreement between dissimilar (*negative*) images. Let $x$ be an image. Assume that a set of positives for $x$ can be acquired, denoted by $\mathcal{X}^+$. Similarly, a set of negatives $\mathcal{X}^-$ is defined. We learn an embedding function $f$ that maps each sample on a normalized hypersphere. The contrastive loss [18, 34] takes the following form

$$\mathcal{L}_{\text{contrastive}} = - \sum_{x^+ \in \mathcal{X}^+} \log \frac{\exp\left[f(x)^T \cdot f(x^+)/\tau\right]}{\exp\left[f(x)^T \cdot f(x^+)/\tau\right] + \sum_{x^- \in \mathcal{X}^-} \exp\left[f(x)^T \cdot f(x^-)/\tau\right]} \quad (1)$$

where $\tau$ is a temperature hyperparameter. We will further refer to the image $x$ as the *anchor*.

SSL methods obtain positives and negatives by treating each image as a separate class [47]. More specifically, augmented views of the same image are considered as positives, while other images are used as negatives. The data augmentation strategy is an important design choice as it determines the invariances that will be learned. Today, most works rely on a similar set of augmentations that consists of (1) cropping, (2) color distortions, (3) horizontal flips and (4) Gaussian blur.

In this paper, we build upon MoCo [19] - a widely known and competitive framework. However, our findings are expected to apply to other related methods (e.g. SimCLR [5]) as well. The embedding function $f$ with parameters $\theta_f$ consists of a backbone $g$ (e.g. ResNet [20]) and a projection MLP

**Table 1: Overview of the training datasets.** We sample a uniform and long-tailed (LT) subset of 118K images from ImageNet. On OpenImages, we sample a random subset of 118K images. The complete train splits are used for COCO and BDD100K. The figure shows some examples.

| Pretrain Data | #Imgs | #Obj/Img | Uniform | Discriminative |
|---|---|---|---|---|
| ImageNet-118K [12] | 118 K | 1.7 | ✓ | ✓ |
| ImageNet-118K-LT [12] | 118 K | 1.7 | ✗ | ✓ |
| COCO [31] | 118 K | 7.3 | ✗ | ✓ |
| OpenImages-118K [28] | 118 K | 8.4 | ✗ | ✓ |
| BDD100K [52] | 90 K | - | ✗ | ✗ |

**Table 2:** Comparison of linear classification models trained on top of frozen features (400 epochs pretraining).

| Pretrain Data | CIFAR10 | Cars | Food | Pets | Places | SUN | VOC |
|---|---|---|---|---|---|---|---|
| IN-118K | 83.1 | 35.9 | 62.2 | 68.9 | 45.0 | 50.0 | 75.8 |
| COCO | 77.4 (−5.7) | 33.9 (−2.0) | 62.0 (−0.2) | 62.6 (−6.3) | 47.3 (+2.3) | 53.6 (+3.6) | 80.9 (+5.1) |
| OI-118K | 74.0 (−9.1) | 32.2 (−3.7) | 58.4 (−3.8) | 59.3 (−9.6) | 46.6 (+1.6) | 52.3 (+2.3) | 75.9 (+0.1) |
| IN-118K-LT | 83.2 (+0.1) | 36.1 (+0.2) | 62.1 (−0.1) | 69.1 (+0.2) | 45.3 (+0.3) | 50.4 (+0.4) | 76.1 (+0.3) |

**Table 3:** Comparison of different representations under the transfer learning setup (400 epochs pretraining).

| | Semantic seg. (mIoU) | | | Detection (AP) | Vid. seg. ($\mathcal{J}\&\mathcal{F}$) | Depth (rmse) |
|---|---|---|---|---|---|---|
| Pretrain Data | VOC | Cityscapes | NYUD | VOC | DAVIS | NYUD |
| IN-118K | 68.9 | 70.1 | 37.7 | 53.0 | 63.5 | 0.625 |
| COCO | 69.1 (+0.2) | 70.3 (+0.2) | 39.3 (+1.6) | 53.0 (+0.0) | 65.1 (+1.6) | 0.612 (−0.013) |
| OI-118K | 67.9 (−1.0) | 70.9 (+0.8) | 38.4 (+0.7) | 53.1 (+0.1) | 64.8 (+1.3) | 0.609 (−0.016) |
| IN-118K-LT | 69.0 (+0.1) | 70.1 (+0.0) | 37.9 (+0.2) | 53.0 (+0.0) | 63.7 (+0.2) | 0.622 (−0.003) |
| BDD100K | - | 70.1 (+0.0) | - | - | - | - |

head $h$. The contrastive loss is applied after the projection head $h$. MoCo uses a queue and a moving-averaged encoder $f'$ to keep a large and consistent set of negative samples. The parameters $\theta_{f'}$ of $f'$ are updated as: $\theta_{f'} = m\theta_{f'} + (1 - m)\theta_f$ with $m$ a momentum hyperparameter. The momentum-averaged encoder $f'$ takes as input the anchor $x$, while the encoder $f$ is responsible for the positives $\mathcal{X}^+$. The queue maintains the encoded anchors as negatives. We refer to [19] for more details.

# 3 Contrastive Learning in the Wild

Most contrastive self-supervised methods train on unlabeled images from ImageNet [12]. This is a curated dataset with unique characteristics. First, the images are *object-centric*, i.e. they depict only a single object in the center of the image. This differs from other datasets [28, 31] which contain more complex scenes with several objects. Second, the underlying classes are *uniformly* distributed. Third, the images have *discriminative* visual features. For example, ImageNet covers various bird species which can be distinguished by a few key features. In contrast, domain-specific datasets (e.g. BDD100K [52]) contain less discriminative scenery showing the same objects like cars, pedestrians, etc. In this section, we study the influence of dataset biases for contrastive self-supervised methods.

**Setup.** We train MoCo-v2 [7] on a variety of datasets. Table 1 shows an overview. The representations are evaluated on six downstream tasks: linear classification, semantic segmentation, object detection, video instance segmentation and depth estimation. We adopt the following target datasets for linear classification: CIFAR10 [27], Food-101 [26], Pets [35], Places365 [56], Stanford Cars [26], SUN397 [48] and VOC 2007 [16]. The semantic segmentation task is evaluated on Cityscapes [10], PASCAL VOC [16] and NYUD [40]. We use PASCAL VOC [16] for object detection. The DAVIS-2017 benchmark [37] is used for video instance segmentation. Finally, depth estimation is performed on NYUD [40]. The model, i.e. a ResNet-50 backbone, is pretrained for 400 epochs using batches of size 256. The initial learning rate is set to 0.3 and decayed using a cosine schedule. We use the default values for the temperature ($\tau = 0.2$) and momentum ($m = 0.999$) hyperparameters.

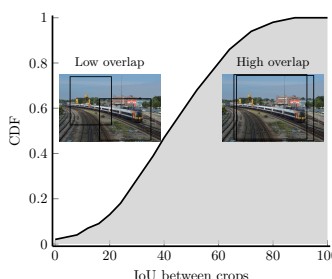

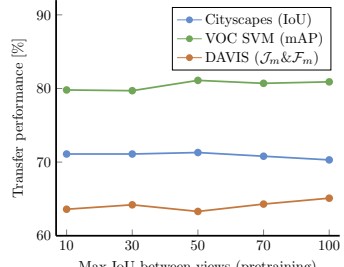

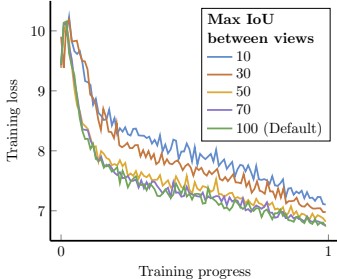

**Figure 1:** IoU between random resized crops for existing frameworks.

**Figure 2:** Transfer results when thresholding the IoU between crops.

**Figure 3:** Training curves when thresholding the IoU between crops.

### 3.1 Object-centric Versus Scene-centric

We compare the representations trained on an object-centric dataset - i.e. ImageNet (IN-118K) - against the ones obtained from two scene-centric datasets - i.e. COCO and OpenImages (OI-118K). Tables 2-3 show the results under the linear classification and transfer learning setup.

**Results.** The linear classification model yields better results on CIFAR10, Cars, Food and Pets when pretraining the representations on ImageNet. Differently, when tackling the classification task on Places, SUN and VOC, the representations from COCO and OpenImages are better suited. The first group of target benchmarks contains images centered around a single object, while the second group contains scene-centric images with multiple objects. We conclude that, for linear classification, the pretraining dataset should match the target dataset in terms of being object- or scene-centric.

Next, we consider finetuning. Perhaps surprisingly, we do not observe any significant disadvantages when using more complex images from COCO or OpenImages for pretraining. In particular, for the reported tasks, the COCO pretrained model even outperforms its ImageNet counterpart. We made a similar observation when increasing the size of the pretraining dataset (see suppl. materials).

**Discussion.** In contrast to prior belief [38, 39], our results indicate that contrastive self-supervised methods do not suffer from pretraining on scene-centric datasets. How can we explain this inconsistency with earlier studies? First, the experimental setup in [38] only considered the linear evaluation protocol for an object-centric dataset (i.e. PASCAL cropped boxes). This analysis [38] does not show us the full picture. Second, the authors conclude that existing methods suffer from using scene-centric datasets due to the augmentation strategy, which involves random cropping. They argue that aggressive cropping could yield non-overlapping views which contain different objects. In this case, maximizing the feature similarity would be detrimental for object recognition tasks. However, the default cropping strategy barely yields non-overlapping views[3]. This is verified by plotting the intersection over union (IoU) between pairs of crops (see Figure 1). We conclude that the used example of non-overlapping crops [38, 39] seldom occurs.

The above observation motivates us to reconsider the importance of using overlapping views. We pretrain on COCO while forcing the IoU between views to be smaller than a predefined threshold. Figures 2-3 show the transfer performance and training curves for different values of the threshold. The optimization objective is harder to satisfy when applying more aggressive cropping (i.e. the training loss increases when lowering the IoU). However, Figure 2 shows that the transfer performance remains stable. Patches within the same image were observed at the same point in time and space, which means that they will share information like the camera viewpoint, color, shape, etc. As a result, the learning signal is still meaningful, even when less overlapping crops are used as positives.

### 3.2 Uniform Versus Long-tailed

Next, we study whether MoCo benefits from using a uniform (IN-118K) versus long-tailed (IN-118K-LT) dataset. We adopt the sampling strategy from [32] to construct a long-tailed version of ImageNet. The classes follow the Pareto distribution with power value $\alpha = 6$. Tables 2-3 indicate that MoCo

---

[3]We use the `RandomResizedCrop` in PyTorch with scaling $s = (0.2, 1.0)$ and output size $224 \times 224$.

**Table 4: Ablation of different components.** Models are pretrained for 200 epochs on COCO using the settings from Section 3. We indicate the differences with MoCo. The best model is trained with additional invariances.

| Method | Setup | | | | | Semantic seg. (mIoU) | | | Classification (mAP / Acc. / Acc.) | | |
|---|---|---|---|---|---|---|---|---|---|---|---|
| | MC | CC | m↓ | A$^+$ | NN | VOC | Cityscapes | NYUD | VOC | ImageNet | Places |
| Rand. init. | - | - | - | - | - | 39.2 | 65.0 | 24.4 | - | - | - |
| MoCo | ✗ | ✗ | ✗ | ✗ | ✗ | 66.2 | 70.3 | 38.2 | 76.1 | 49.3 | 45.1 |
| Sec. 4.1 | ✓ | ✗ | ✗ | ✗ | ✗ | 69.9 (+3.7) | 70.9 (+0.6) | 39.4 (+1.2) | 81.3 (+5.2) | 53.4 (+4.1) | 47.7 (+2.6) |
| | ✓ | ✓ | ✗ | ✗ | ✗ | 70.2 (+4.0) | 70.9 (+0.6) | 39.5 (+1.3) | 82.1 (+6.0) | 54.0 (+4.7) | 47.9 (+2.8) |
| | ✓ | ✓ | ✓ | ✗ | ✗ | 70.9 (+4.7) | 71.3 (+1.0) | 39.9 (+1.7) | 82.8 (+6.7) | 54.8 (+5.5) | 48.1 (+3.0) |
| Sec. 4.2 | ✓ | ✓ | ✓ | ✓ | ✗ | 71.4 (+5.2) | 72.0 (+1.7) | 40.0 (+1.8) | 83.7 (+7.6) | 55.5 (+6.2) | 48.2 (+3.1) |
| Sec. 4.3 | ✓ | ✓ | ✓ | ✓ | ✓ | 71.9 (+5.7) | 72.2 (+1.9) | 40.9 (+2.7) | 85.1 (+9.0) | 55.9 (+6.6) | 48.5 (+3.4) |

MC: Multi-crop, CC: Constrained multi-crop, m↓: Lower momentum, A$^+$: Stronger augmentations, NN: Nearest neighbors

is robust to changes in the class distribution of the dataset. In particular, the IN-118K-LT model performs on par or better compared to its IN-118K counterpart across all tasks. We conclude that it is not essential to use a uniformly distributed dataset for pretraining.

### 3.3 Domain-Specific Datasets

Images from ImageNet have discriminative visual features, e.g. the included bird species can be recognized from their beak or plumage. In this case, the model could achieve high performance on the instance discrimination task by solely focusing on the most discriminative feature in the image. Differently, in urban scene understanding, we deal with more monotonous scenery - i.e. all images contain parked cars, lane markings, pedestrians, etc. We try to measure the impact of using less discriminative images for representation learning.

We pretrain on BDD100K - a dataset for urban scene understanding. Table 3 compares the representations for the semantic segmentation task on Cityscapes when pretraining on BDD100K versus IN-118K. The BDD100K model performs on par with the IN-118K pretrained baseline (BDD100K: $+0.00\%$). This shows that MoCo can be applied to domain-specific data as well. Two models trained on very different types of data perform equally well. A recent study [54] showed that it is mostly the low- and mid-level visual features that are retained under the transfer learning setup. This effect explains how two different representations produce similar results. Finally, we expect that further gains can be achieved by finetuning the augmentation strategy using domain knowledge.

## 4 Learning Invariances

In Section 3, we showed that MoCo works well for a large variety of pretraining datasets: scene-centric, long-tailed and domain-specific images. This result shows that there is no obvious need to use a more advanced pretext task [39] or use an alternative source of data like video [38] to improve the representations. Therefore, instead of developing a new framework, we try to realize further gains while sticking with the approach from MoCo. We realize this objective by learning additional invariances. We study three such mechanisms, i.e. multi-scale constrained cropping, stronger augmentations and the use of nearest neighbors. We concentrate on the implementation details and uncover several interesting qualities of the learned representations. Table 4 shows an overview of the results.

### 4.1 Multi-Scale Constrained Cropping

The employed cropping strategy proves crucial to obtain powerful visual representations. So far, we used the default two-crop transform, which samples two image patches to produce an anchor and a positive. The analysis in Section 3 showed that this transformation yields highly-overlapping views. As a result, the model can match the anchor with its positive by attending only to the most discriminative image component, while ignoring other possibly relevant regions. This observation leads us to revisit the `multi-crop` augmentation strategy from [4].

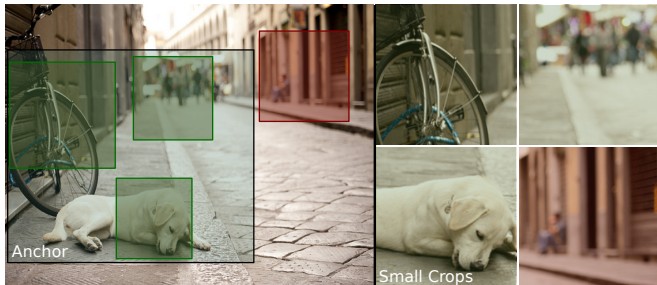 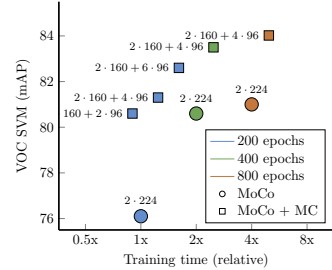

**Figure 4:** The constrained multi-crop. Smaller crops are forced to lie within the anchor image. Invalid/valid crops are colored in red/green.

**Figure 5:** Transfer performance versus time for the multi-crop (MC) model.

**Setup.** The `multi-crop` transform samples $N$ additional positive views for each anchor. Remember, the views are generated by randomly applying resized cropping, color distortions, horizontal flipping and Gaussian blur. We adjust the scaling parameters and resolution of the random crop transform to limit the computational overhead. The two default crops (i.e. the anchor and a positive) contain between $20\%$ and $100\%$ of the original image area, and are resized to $160 \times 160$ pixels. Differently, the $N$ additional views contain between $5\%$ and $14\%$ of the original image area, and are resized to $96 \times 96$ pixels. We maximize the agreement between all $N + 1$ positives and the anchor. Note that the $N$ smaller, more aggressively scaled crops are not used as anchors, i.e. they are not shown to the momentum-updated encoder and are not used as negatives in the memory bank.

Further, we consider two additional modifications. First, we enforce the smaller crops to overlap with the anchor image. In this way, all positives can be matched with their anchor as there is shared information. The new transformation is referred to as `constrained multi-cropping`. Figure 4 illustrates the concept. Second, since increasing the number of views facilitates faster training, we reduce the momentum hyperparameter $m$ from 0.999 to 0.995. We pretrain for 200 epochs on COCO.

**Results.** Table 4 benchmarks the modifications. Both the `multi-crop` (MC) and `constrained multi-crop` (CC) significantly improve the transfer performance. Reducing the momentum (m↓) yields further gains. Figure 5 plots the training time versus performance for the `multi-crop` model. The `multi-crop` improves the computational efficiency w.r.t. the two-crop transform.

**Discussion.** We investigate what renders the `multi-crop` and `constrained multi-crop` effective. The network matches the representations of the large anchor image with the small crops. Since the small crops show random subparts of the anchor image, the network is forced to encode as much information as possible, such that all image regions can be retrieved from the representation of the scene. As a result, the representations will be informative of the spatial layout and different objects in the image - rather than attend only to the most discriminative image component.

The latter is verified by visualizing the class activation maps (CAMs) [55] of different representations (see Figure 6). The CAMs of the `multi-crop` model segment the complete object, while the `two-crop` model only looks at a few discriminative components. Additional examples can be found in the suppl. materials.

**Table 5: DAVIS 2017 video instance segmentation**. We use the publicly available code from [24] to evaluate our frozen representations. Qualitative results are shown for MoCo trained with the `multi-crop` (MC) transform.

| Method | Data | $\mathcal{J}_m \uparrow$ | $\mathcal{F}_m \uparrow$ |
|---|---|---|---|
| DenseCL [45] | COCO | 60.6 | 63.9 |
| MAST [29] | YT-VOS | 63.3 | 67.6 |
| STC [24] | Kinetics | 64.8 | 70.2 |
| MoCo | COCO | 61.6 | 66.6 |
| MoCo + MC | COCO | 64.3 | 69.4 |

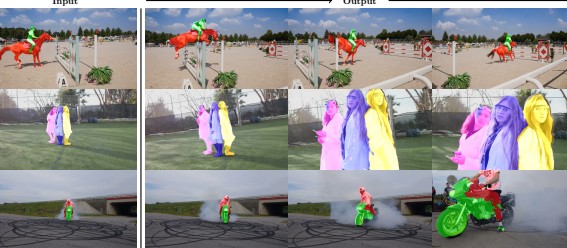

**Table 6: VOC semantic segment retrieval.** Each image is partitioned into a number of segments by running K-Means (K=15) on the spatial features. Then we adopt a region descriptor - computed by averaging the embeddings of all pixels within a segment - to obtain nearest neighbors of the validation regions from the train set. Qualitative results are shown for MoCo trained with the `multi-crop` (MC) augmentation strategy (K=5).

| Method | Segm. Retr. VOC | |
| --- | --- | --- |
| | 7 classes (IoU)↑ | 21 classes (IoU)↑ |
| SegSort [53] | 10.2 | - |
| SSL HG [53] | 24.6 | - |
| DenseCL [45] | 48.4 | 35.1 |
| MoCo | 41.8 | 28.1 |
| MoCo + MC | 48.1 | 35.1 |

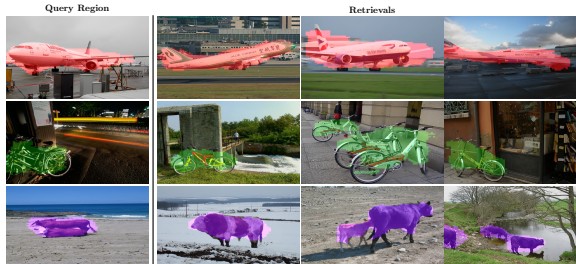

Further, the representations are evaluated on two dense prediction tasks without any finetuning. We consider the tasks of video instance segmentation on DAVIS 2017 [37] and semantic segment retrieval on VOC [16]. The `multi-crop` model is found superior at modeling dense correspondences for the video instance segmentation task in Table 5 (MoCo vs. MoCo + MC). Similarly, Table 6 shows that the `multi-crop` model achieves higher performance on the semantic segment retrieval task compared to its two-crop counterpart. This indicates that the pixel embeddings are better disentangled according to the semantic classes. Finally, the MoCo `multi-crop` model is competitive with other methods that were specifically designed for the tasks. We conclude that the `multi-crop` setup provides a viable alternative to learn dense representations without supervision. Moreover, this setup does not rely on video [24, 29] or handcrafted priors [23, 44, 53].

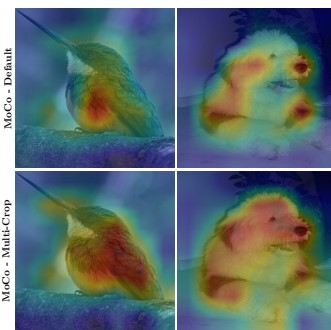

**Figure 6:** Class activation maps for the two-crop vs. multi-crop model.

## 4.2 Stronger Augmentations

As mentioned before, the image crops are augmented by random color distortions, horizontal flips and Gaussian blur. Can we do better by using stronger augmentations? We investigate the use of AutoAugment [11] - an advanced augmentation policy obtained using supervised learning on ImageNet. We consider three possible strategies to augment the positives: (1) standard augmentations, (2) AutoAugment and (3) randomly applying either (1) or (2).

Table 7 compares the representations under the linear evaluation protocol on PASCAL VOC. Replacing the standard augmentations with AutoAugment degrades the performance (from 82.8% to 79.9%). However, randomly applying either the standard augmentations or AutoAugment does improve the result (from 82.8% to 83.7%). Chen *et al.* [5] showed that contrastive SSL benefits from using strong color augmentations as two random crops from the same image will share a similar color distribution. AutoAugment applies fewer color distortions, resulting in lower performance. This is

**Table 7:** Ablation of augmentation policies applied to random crops.

| Augmentation policy | VOC SVM (mAP) |
| --- | --- |
| Standard | 82.8 |
| AutoAugment | 79.9 |
| Standard or AutoAugment | 83.7 |

compensated when combining both augmentation strategies. Finally, Table 4 shows that combining our custom augmentation policy ($A^+$) with the model from Section 4.1 results in considerable improvements on all tasks.

## 4.3 Nearest Neighbors

Prior work [43] showed that the model learns to map visually similar images closer together than dissimilar ones when tackling the instance discrimination task. In this section, we build upon this insight by imposing additional invariances between neighboring samples. By leveraging other samples as positives, we can capture a rich set of deformations that are hard to model via handcrafted augmentations. However, a well-known problem with methods that group different samples together

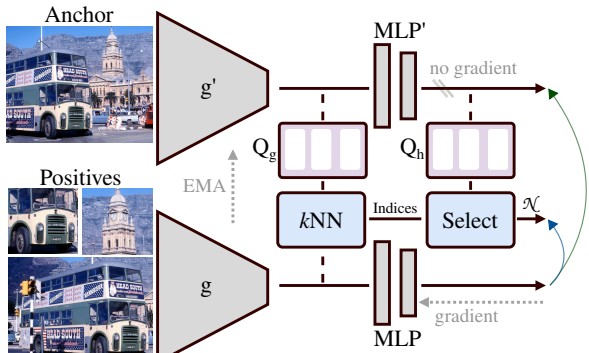

**Figure 7: kNN-MoCo setup.** $Q_g$ and $Q_h$ maintain aligned queues of backbone and output features. As before, the encoder $f$ needs to match positives with their anchor (green arrow). Additionally, we also match the positives with their $k$ nearest neighbors obtained from the queue $Q_g$ (blue arrow).

**Algorithm 1** Pseudocode for kNN-MoCo

```
# g, g': the (momentum-updated) backbone
# h, h': the (momentum-updated) proj. head
# x: batch of anchor images  # B
# x+: batch of positives     # B·L
# Q_g: queue of x_g's         # C_g xK
# Q_h: queue of x_h's         # C_h xK

x_g,x_g+= g'(x),g(x+)  # BxC_g, B·LxC_g
x_h,x_h+= h'(x_g),h(x_g+)  # BxC_h, B·LxC_h
x_g,x_h= x_g.detach(), x_h.detach()

l_pos= bmm(x_h+.view(B,L,C_h),x_h.view(B,C_h,1))
l_neg= mm(x_h+,Q_h).view(B,L,K)
logits = cat([l_pos, l_neg],dim=2) # BxLx(K+1)
indices = topk(mm(x_g+,Q_g).view(B,L,K),dim=2)

# loss: (1) sharpen with t, (2) apply CE
loss_inst = CE(logits/t, zeros((B,L)))
loss_nn = multi_label_CE(l_neg/t, indices)
loss_tot = loss_inst + lambda * loss_nn
```

bmm: batch matrix mult.; mm: matrix mult.; cat: concatenate; topk: $k$ indices of top $k$ elements; CE: cross-entropy

---

is preventing representation collapse. Therefore, we formulate our intuition as an auxiliary loss that regularizes the representations - keeping the instance discrimination task as the main force.

**Setup.** Recall that the encoder $f$ consists of a backbone $g$ and a projection head $h$. MoCo maintains a queue $Q_h$ of encoded anchors $\{q_0^h \dots q_{K-1}^h\}$ processed by the momentum encoder $f'$. We now introduce a second - equally sized and aligned - queue $Q_g$ which maintains the features from before the projection head $\{q_0^g \dots q_{K-1}^g\}$. The queue of backbone features $Q_g$ can be used to mine nearest neighbors on-the-fly during training. In particular, for a positive $x^+$ - processed by the encoder $f$ - the set of its $k$ nearest neighbors $\mathcal{N}_{x^+} = \{q_i^h \mid sim(g(x^+), q_i^g) \text{ is top } k \in Q_g\}$ is computed w.r.t. the queue $Q_g$. The cosine similarity measure is denoted by $sim$. Finally, we use the contrastive loss to maximize the agreement between $x^+$ and its nearest neighbors $\mathcal{N}_{x^+}$ after the projection head:

$$\mathcal{L}_{\text{nn}} = - \sum_{x^+ \in \mathcal{X}^+} \frac{1}{k} \sum_{q \in \mathcal{N}_{x^+}} \log \frac{\exp\left[q^T \cdot f(x^+)/\tau\right]}{\exp\left[q^T \cdot f(x^+)/\tau\right] + \sum_{x^- \in \mathcal{X}^-} \exp\left[q^T \cdot f(x^-)/\tau\right]}. \quad (2)$$

The total loss is the sum of the instance discrimination loss $\mathcal{L}_{\text{inst}}$ and the nearest neighbors loss $\mathcal{L}_{\text{nn}}$ from Eq. 2: $\mathcal{L}_{\text{inst}} + \lambda\mathcal{L}_{\text{nn}}$. Figure 7 shows a schematic overview of our k-Nearest Neighbors based Momentum Contrast setup (kNN-MoCo). Algorithm 1 contains the pseudocode (see also suppl.).

**Table 8:** Ablation study of the number of neighbors $k$ and weight $\lambda$ for a linear classifier on VOC. $\lambda = 0$ represents the multi-crop model from Section 4.1. Models are trained for 200 epochs on COCO.

| $k$ | 1 | 5 | 10 | 20 | 50 | | $\lambda$ | 0.0 | 0.1 | 0.2 | 0.4 | 0.8 |
|---|---|---|---|---|---|---|---|---|---|---|---|---|
| mAP (%) | 83.6 | 84.0 | 84.1 | 84.2 | 84.3 | | mAP (%) | 82.8 | 83.8 | 84.1 | 84.2 | 80.7 |

**Table 9: State-of-the-art comparison.** MoCo and DenseCL are trained for 800 epochs on COCO. VirTex is trained on COCO captions [8]. MoCo is trained while imposing various additional invariances.

| | Semantic seg. (mIoU) | | | Classification (mAP / Acc. / Acc.) | | | Vid. Seg. ($\mathcal{J}\&\mathcal{F}$) | Depth (rmse) |
|---|---|---|---|---|---|---|---|---|
| **Method** | **VOC** | **Cityscapes** | **NYUD** | **VOC** | **ImageNet** | **Places** | **DAVIS** | **NYUD** |
| Rand. init. | 39.2 | 65.0 | 24.4 | - | - | - | 40.8 | 0.867 |
| DenseCL [45] | 73.2 | **73.5** | **42.1** | 83.5 | 49.9 | 45.8 | 61.8 | 0.589 |
| VirTex [13] | 72.7 | 72.5 | 40.3 | **87.4** | 53.8 | 40.8 | 61.3 | 0.613 |
| MoCo | 71.1 | 71.3 | 40.0 | 81.0 | 49.8 | 44.7 | 63.3 | 0.606 |
| + CC | 72.2 | 71.6 | 40.4 | 84.0 | 54.6 | 46.1 | 65.5 | 0.595 |
| + CC + A$^+$ | 72.7 | 71.8 | 40.7 | 85.0 | 56.0 | 47.0 | 65.7 | 0.590 |
| + CC + A$^+$ + kNN | **73.5** | 72.3 | 41.3 | 85.9 | **56.1** | **48.6** | **66.2** | **0.580** |

CC: Constrained multi-crop (Sec. 4.1), A$^+$: Stronger augmentations (Sec. 4.2), kNN: nearest neighbors (Sec. 4.3).

**Results & Discussion.** Table 8 contains an ablation study of the number of neighbors $k$ and the weight $\lambda$. The performance remains stable for a large range of neighbors $k$ ($\lambda$ is fixed at 0.4). However, increasing the number of neighbors $k$ positively impacts the accuracy. We use $k = 20$ for the remainder of our experiments. Further, the representation quality degrades when using a large weight (e.g. $\lambda = 0.8$). This shows the importance of using the instance discrimination task as our main objective. Also, not shown in the table, we found that it is important to mine the neighbors using the features before the projection head (84.2% vs 82.8%). Finally, Table 4 shows improved results on all tasks when combining the nearest neighbors loss with our other modifications. In conclusion, we have successfully explored the data manifold to learn additional invariances. The proposed implementation can be seen as a simple alternative to clustering-based methods [2, 4, 30, 46].

## 4.4 Discussion

We retrain our final model for 800 epochs on COCO and compare with two other methods, i.e. DenseCL [45] and Virtex [13]. We draw the following conclusions from the results in Table 9. First, our model improves over the MoCo baseline. The proposed modifications force the model to learn useful features that can not be learned through standard data augmentations, even when increasing the training time. Second, our representations outperform other works on several downstream tasks. These frameworks used more advanced schemes [45] or caption annotations [13]. Interestingly, DenseCL reports better results for the segmentation tasks on Cityscapes and NYUD, but performs worse on other tasks. In contrast, the performance of our representations is better balanced across tasks. We conclude that generic pretraining is still an unsolved problem.

## 5 Related Work

**Contrastive learning.** The idea of contrastive learning [18, 34] is to attract positive sample pairs and repel negative sample pairs. Self-supervised methods [5, 9, 17, 19, 33, 41, 42, 47, 51] have used the contrastive loss to learn visual representations from unlabeled images. Augmentations of the same image are used as positives, while other images are considered as negatives.

A number of extensions were proposed to boost the performance. For example, a group of works [21, 36, 44, 45] applied the contrastive loss at the pixel-level to learn dense representations. Others improved the representations for object recognition tasks by re-identifying patches [50] or by maximizing the similarity of corresponding image regions in the intermediate network layers [49]. Finally, Selvaraju *et al.* [39] employed an attention mechanism to improve the visual grounding abilities of the model. In contrast to these works, we do not employ a more advanced pretext task to learn spatially structured representations. Instead, we adopt a standard framework [19] and find that the learned representations exhibit similar properties when modifying the cropping strategy. Further, we expect that our findings can be relevant for other contrastive learning frameworks too.

**Clustering.** Several works combined clustering with self-labeling [2, 3] or contrastive learning [4, 30, 46] to learn representations in a self-supervised way. Similar to the nearest neighbors loss (Eq. 2), these frameworks explore the data manifold to learn invariances. Differently, we avoid the use of a clustering criterion like K-Means by computing nearest neighbors on-the-fly w.r.t. a memory bank. A few other works [22, 43] also used nearest neighbors as positive pairs in an auxiliary loss. However, the neighbors had to be computed off-line at fixed intervals during training. Concurrent to our work, Dwibedi *et al.* [14] adopted nearest neighbors from a memory bank under the BYOL [17] framework. The authors focus on image classification datasets. In conclusion, we propose a simple, yet effective alternative to existing clustering methods.

**Other.** Contrastive SSL has been the subject of several recent surveys [15, 38, 54]. We list the most relevant ones. Similar to our work, Zhao *et al.* [54] pretrain on multiple datasets. They investigate what information is retained under the transfer learning setup, which differs from the focus of this paper. Purushwalkam and Gupta [38] study the influence of the object-centric dataset bias, but their experimental scope is rather limited, and their conclusions diverge from the ones in this work. Ericsson *et al.* [15] compare several ImageNet pretrained models under the transfer learning setup. In conclusion, we believe our study can complement these works.

# 6 Conclusion

In this paper, we showed that we can find a generic set of augmentations/invariances that allows us to learn effective representations across different types of datasets (i.e., scene-centric, non-uniform, domain-specific). We provide empirical evidence to support this claim. First, in Section 3, we show that the standard SimCLR augmentations can be applied across several datasets. Then, Section 4 studies the use of additional invariances to improve the results for a generic dataset (i.e., MS-COCO). In this way, we reduce the need for dataset or domain-specific expertise to learn useful representations in a self-supervised way. Instead, the results show that simple contrastive frameworks apply to a wide range of datasets. We believe this is an encouraging result. Finally, our overall conclusion differs from a few recent works [38, 39], which investigated the use of a more advanced pretext task or video to learn visual representations.

Our paper also yields a few interesting follow-up questions. **Modalities.** Can we reach similar conclusions for other modalities like video, text, audio, etc.? **Invariances.** What other invariances can be applied? How do we bias the representations to focus more on texture, shape or other specific properties? **Compositionality.** Can we combine different datasets to learn better representations?

## Broader Impact

The goal of this work is to study and improve contrastive self-supervised methods for learning visual representations. Self-supervised learning aims to learn useful representations without relying on human annotations. Our analysis indicates that existing methods can be applied to a large variety of datasets. This observation could benefit applications where annotated data is scarce, like medical imaging, or where large amounts of unlabeled data are readily available, like autonomous driving. Our work also improves the learned representations, and thus benefits many downstream tasks like semantic segmentation, classification, etc. These tasks are of relevance to many applications. At this point, it is hard to assess all possible societal implications of this work. After all, the advantages or disadvantages of new applications using the studied technology will depend on the intentions of the users or inventors.

**Acknowledgment.** The authors thankfully acknowledge support by Toyota via the TRACE project and MACCHINA (KU Leuven, C14/18/065). This work is also sponsored by the Flemish Government under the Flemish AI programme.

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
