# Supplementary Materials

**Wouter Van Gansbeke**[*,1]    **Simon Vandenhende**[*,1]    **Stamatios Georgoulis**[2]    **Luc Van Gool**[1,2]

[1] KU Leuven/ESAT-PSI    [2] ETH Zurich/CVL

## A   Implementation Details

This section provides all necessary details to reproduce the results from the main paper. The code will be made available upon acceptance. All pretraining experiments were run with $2 \times 32$GB V100 GPUs.

### A.1   Pretraining

**Datasets.**   Four different datasets are used for pretraining. These include MS-COCO [18], ImageNet [8], OpenImages [17] and BDD100K [31]. All datasets are publicly available and free to use for research purposes. Images from the official train splits are used for pretraining.

We construct the IN-118K dataset by sampling a uniform subset from the ImageNet train split. Differently, the IN-118K-LT is a long-tailed version of the ImageNet train split. The classes follow the Pareto distribution with power value $\alpha = 6$ (see Figure S1). OI-118K is obtained by randomly sampling 118K images from the OpenImages-v4 train split. Note that all datasets are constructed to be of similar size, which facilitates an apples-to-apples comparison.

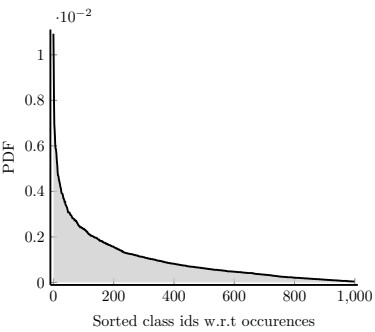

**Figure S1:** Class distribution of the IN-118K-LT dataset.

**MoCo.**   The official implementation of MoCo-v2 [5] is used to run our experiments. The model consists of a ResNet-50 backbone with MLP head. We do not include batchnorm in the head. The models in Section 3 and 4 are pretrained for 400 and 200 epochs respectively. The initial learning rate is set to 0.3. Other hyperparameters use the default values.

**Multi-Crop.**   We modify the MoCo framework to support multi-crop training. In particular, $N$ additional smaller crops are used as positives for the anchor. Note that the crops are not processed by the momentum-encoder and do not appear as negatives in the memory bank. The crops are generated with the `RandomResizedCrop` transform from PyTorch [22]. The scale is selected similar to [2]. The two large crops - the anchor and the positive - are obtained with the scale randomly selected between $[0.2, 1.0]$ and output size $160 \times 160$ pixels. Differently, we randomly sample the scale between $[0.05, 0.14]$ and use output size $96 \times 96$ pixels for the $N$ smaller crops. The `constrained multi-crop` is implemented by checking whether the smaller crops overlap with the anchor crop. The smaller crops are resampled if less than 20% of their area overlaps with the anchor crop.

**Stronger Augmentations.**   The dataloader is modified to support stronger augmentations. As before, we first apply the `RandomResizedCrop` transform to the input image. This is followed by either (1) the remaining default transformations or (2) AutoAugment [7]. For AutoAugment, we use the official PyTorch implementation. The augmentation policy was obtained from ImageNet [8]. Since the use of stronger augmentations only impacts the input data, the rest of the setup is kept

35th Conference on Neural Information Processing Systems (NeurIPS 2021).

unchanged. Note that we only apply the stronger augmentations to the positives. The anchors are augmented using the default transformations.

**Nearest Neighbors.** The forward pass of the model is modified to incorporate the auxiliary loss. Other settings are kept. A PyTorch-like implementation of our algorithm can be found in Section B. The auxiliary loss uses weight $\lambda = 0.4$ and 20 nearest neighbors.

## A.2 Linear classification

We train a linear classifier on top of the pretrained ResNet-50 backbone. The weights and batchnorm statistics of the backbone are frozen. The top-1 accuracy metric is reported for CIFAR-10 [16], Stanford Cars [15], Food-101 [1], Pets [21], Places [33], SUN397 [30] and ImageNet [8]. We report the mAP metric on PASCAL VOC 2007 [11]. SUN397 uses the first train/test split, while other datasets use the regular train/val/test splits. All datasets are publicly available and free to use for research purposes.

On ImageNet and Places, we adopt the evaluation protocol from He *et al.* [13]. The training follows the typical ResNet example in PyTorch [22]. Standard data augmentations are used. The model is trained for 100 epochs using SGD with momentum 0.9 and initial learning rate 30.0. The linear layer is regularized with weight decay 0.0001. The learning rate is decayed with a factor 10 after 60 and 80 epochs.

The other datasets are evaluated following the protocol from Ericsson *et al.* [10]. We fit a multinomial logistic regression model to the features extracted from the backbone. The implementation from `sklearn` is used. We do not apply any augmentations, and resize all images to $224 \times 224$ pixels using the `Resize` transform in PyTorch. The L2 regularisation constant is selected on the validation set using 45 logarithmically spaced values between 1e-5 and 1e5. We use the L-BFGS optimizer and softmax cross-entropy objective. On PASCAL VOC, we need to solve a multi-class classification problem. In this case, we fit a binary classifier for each class.

## A.3 Segmentation

The representations are finetuned end-to-end for the semantic segmentation task on Cityscapes [6], PASCAL VOC [11] and NYUD [25]. All datasets are publicly available and free to use for research purposes. On Cityscapes, we train on the `train_fine` set (2975 images) and evaluate on the `val` set. For PASCAL VOC, training is performed on the `train_aug2012` set (10582 images) and evaluation on the `val2012` set. Finally, the `train` set (795 images) and `val` set are used for training and evaluation on NYUD. We adopt the mean intersection over union (mIoU) as evaluation metric. The training images are augmented with random scaling (ratio between 0.5 and 2.0) and horizontal flipping.

**PASCAL VOC.** We follow the evaluation protocol from [13] and use an FCN-based [19] model. The $3 \times 3$ convolutions in the final ResNet-block have dilation 2 and stride 1. The backbone is followed by two additional $3 \times 3$ convolutions of 256 channels with BN and ReLU, and then a $1 \times 1$ convolution to obtain pixel-wise predictions. We use dilation rate 6 in the extra $3 \times 3$ convolutions. The model is trained for 45 epochs using batches of size 16. We use SGD with momentum 0.9 and initial learning rate 0.003. The learning rate is multiplied with 0.1 after 30 and 40 epochs. Weight decay regularization is used of 0.0001. We use random crops of size $513 \times 513$ pixels during training.

**Cityscapes.** The setup from PASCAL VOC is reused. The dilation rate of the $3 \times 3$ convolutions in the head is set to 1. This improves the results on Cityscapes, since the model can better capture small objects and classes with a thin structure like poles. The model is trained for 150 epochs using batches of size 16. We use the Adam optimizer with initial learning rate 0.0001 and a poly learning rate schedule. Weight decay regularization is used of 0.0001. Training uses random crops of size $768 \times 768$ pixels.

**NYUD.** The publicly available code from [27] is used to evaluate the representations on NYUD. Again, the $3 \times 3$ convolutions in the final ResNet-block have dilation 2 and stride 1. The backbone is followed by a DeepLab-v3 decoder [3]. The model is trained for 100 epochs using batches of size 8. We use the Adam optimizer with initial learning rate 0.0001 and a poly learning rate schedule.

Weight decay regularization is used of 0.0001. We use random crops of size $512 \times 512$ pixels during training.

### A.4  Object Detection

The representations are evaluated for the `VOC` 2007 object detection task [11]. The default metrics are reported on the 2007 `test` split. We adopt the evaluation protocol from He *et al.* [13]. The code uses the `Detectron2` framework [29]. The detector is Faster R-CNN [24] with R50-dilated-C5 backbone. All layers are finetuned end-to-end. The image scale is [480, 800] pixels during training and 800 at inference.

### A.5  Depth Estimation

We evaluate the representations for the depth estimation task on NYUD [25]. The model and training setup are the same as for the semantic segmentation task on NYUD. We adopt the mean squared error objective function to train the model. The results report the root mean squared error (rmse) on the validation set.

### A.6  Video Instance Segmentation

The DAVIS-2017 video instance segmentation benchmark [23] is used. The dataset is publicly available and free to use for research purposes. The mean region similarity $\mathcal{J}_m$ and mean contour-based accuracy $\mathcal{F}_m$ are reported on the 2017 `val` split. We adopt the evaluation protocol from Jabri *et al.* [14]. The labels are propagated using nearest neighbors between consecutive frames. Note that we do not apply any finetuning for the task.

### A.7  Semantic Segment Retrieval

Finally, we evaluate the representations for the semantic segment retrieval task on PASCAL VOC [11]. The train and val splits are the same as for the semantic segmentation task. The model is a ResNet-50. The $3 \times 3$ convolutions in the final ResNet-block use dilation 2 and stride 1. The average pooling layer is removed to maintain the spatial structure of the output. The input images are rescaled to $512 \times 512$ pixels and the output is of size $64 \times 64$ For each image, we cluster the spatial features with K-means (K=15). This results in K regions per image. For each region, we compute a region descriptor by averaging the features of all pixels within the region. The validation regions are assigned a label by obtaining the nearest neighbor from the train set. Note that we do not use a weighted k-NN algorithm. The predictions are rescaled to match the original input resolution for both evaluation and visualization. The mean intersection over union (mIoU) is reported on the validation set.

## B  Pseudocode

Algorithm 1 provides the pseudocode of our framework. The code supports the `constrained multi-crop`, stronger augmentations and the nearest neighbors auxiliary loss. The implementation builds on top of MoCo. However, the ideas in this paper could be combined with other frameworks too. We highlight the most important differences with MoCo. In order to mine nearest neighbors that are visually similar, the representations should already model semantically meaningful properties. However, at the start of training, the model is initialized randomly. In this case, we can not learn any useful invariances from grouping nearest neighbors. We resolve this problem by only applying the auxiliary loss after a fixed number of epochs (5).

We found an alternative implementation which allows to adopt the auxiliary loss from the start of training. We can update the memorybank (final line in Algorithm 1) before computing the loss. In this case, there is a mismatch between the logits and the memory bank for the most recent batch [1]. This means we will inject random noise when the nearest neighbors map to the most recently used batch in the memory bank. We observe that this behavior mostly occurs early on during training, as

---

[1]The memory bank is implemented as an array of fixed size. We replace the oldest batch when adding a new batch using a pointer.

the memory bank is initialized randomly. Over time, this effect is reduced, and we end up using a loss that is similar to what we had before. We report similar results for both strategies.

---

**Algorithm 1** Pseudocode of kNN-MoCo in a PyTorch-like style.

```
# g, g_m: the backbone g and momentum updated backbone g_m
# h, h_m: the projection head h and momentum updated projection head h_m (MLP)
# anchors: batch of anchor images (B x 3 x 224 x 224)
# positives: batch of global positive views (B x 3 x 224 x 224)
# positives_small: batch of N local positive views (B*N x 3 x 96 x 96)
# queue: dictionary as a queue (C x K)
# queue_g: dictionary as a queue (C_g x K)
# m: momentum
# t: temperature
# k: number of nearest neighbors
# lambda: weight in range [0, 1]

g_m.params = g.params # initialize momentum updated backbone
h_m.params = h.params # initialize momentum updated head

for batch in loader: # load a minibatch with B samples

    # randomly augment batch
    anchors   = aug(batch)
    positives = aug(batch)
    positives_small = aug_small(batch, anchors) # constrained multi-crop      Sec. 4.1

    # forward pass
    anchors_g, positives_g = g_m(anchors), g(positives)     # B x C_g
    anchors, positives = h_m(anchors_g), h(positives_g)     # B x C
    positives_small_g = g(positives_small)                  # B*N x C_g
    positives_small = h(positives_small_g)                  # B*N x C

    # concatenate positive views
    positives_g = cat([positives_g, positives_small_g], dim=0)  # B*(N+1) x C_g
    positives = cat([positives, positives_small], dim=0)        # B*(N+1) x C

    # compute logits
    l_pos = bmm(positives.view(B,N+1,C), anchors.view(B,C,1))   # B x N+1 x 1
    l_neg = mm(positives, queue.view(C,K)).view(B,N+1,K)        # B x N+1 x K
    logits = cat([l_pos, l_neg], dim=2)                         # B x N+1 x K+1

    # determine indices of the nearest neighbors
    indices = topk(mm(positives_g, queue_g).view(B,N+1,K),dim=2)# B x N+1 x k   Sec. 4.3

    # loss: (1) sharpen with temperature t, (2) apply cross-entropy loss
    loss_inst = CE(logits/t, zeros((B,N+1)))
    loss_nn = multi_label_CE(l_neg/t, indices)
    loss = loss_inst + lambda * loss_nn

    # SGD update: g, h
    loss.backward()
    update(g.params, h.params)

    # momentum update: g_m, h_m
    g_m.params = m * g_m.params + (1-m) * g.params
    h_m.params = m * h_m.params + (1-m) * h.params

    # update dictionaries: queue & queue_g
    enqueue_dequeue(queue, anchors)
    enqueue_dequeue(queue_g, anchors_g)
```

bmm: batch matrix mult.; mm: matrix mult.; cat: concatenate; topk: indices of top-k largest elements; CE: cross-entropy loss
Important differences with MoCo are highlighted.

## C   Additional Pretraining Results

The experiments in the main paper were mostly performed on datasets of moderate size (e.g. MS-COCO). In this section, we include additional results when pretraining on datasets of larger size. Again, we compare the use of object-centric (ImageNet) versus scene-centric (OpenImages) images for pretraining. We start from the IN-118K and OI-118K datasets defined in the main paper, and increase the size of the dataset by a factor of 2, 4 and 8. The representations are again evaluated by transferring them to multiple dense prediction tasks.

Table S1 shows the results. We draw the following two conclusions. First, as expected, the numbers improve when more data is used. However, the results show diminishing returns when scaling up the dataset size. He *et al.* made a similar observation. Second, there are no significant disadvantages to

**Table S1:** Comparison of representations trained on datasets of varying size. Results are obtained when applying end-to-end finetuning. We indicate the differences with the $1\times$ baseline.

| Pretrain dataset | Size | Factor | Semantic seg. (mIoU) | | | Vid. seg. ($\mathcal{J}\&\mathcal{F}$) | Depth (rmse) |
|---|---|---|---|---|---|---|---|
| | | | **VOC** | **Cityscapes** | **NYUD** | **DAVIS** | **NYUD** |
| IN-118K | 118K | $1\times$ | 68.9 | 70.1 | 37.7 | 63.5 | 0.625 |
| | 236K | $2\times$ | 72.4 (+3.5) | 71.1 (+1.0) | 39.9 (+2.2) | 64.9 (+1.4) | 0.612 (−0.013) |
| | 472K | $4\times$ | 73.5 (+4.6) | 71.1 (+1.0) | 39.9 (+2.2) | 64.9 (+1.4) | 0.607 (−0.018) |
| | 944K | $8\times$ | 75.0 (+6.1) | 71.7 (+1.6) | 40.9 (+3.2) | 66.1 (+2.6) | 0.599 (−0.026) |
| OI-118K | 118K | $1\times$ | 67.9 | 70.9 | 38.4 | 64.8 | 0.609 |
| | 236K | $2\times$ | 71.4 (+3.5) | 71.4 (+0.5) | 40.1 (+1.7) | 64.5 (−0.3) | 0.609 ( 0.0) |
| | 472K | $4\times$ | 72.6 (+4.7) | 71.9 (+1.0) | 40.9 (+2.5) | 64.2 (−0.6) | 0.601 (−0.008) |
| | 944K | $8\times$ | 73.6 (+5.7) | 72.2 (+1.3) | 41.0 (+2.6) | 65.0 (+0.2) | 0.600 (−0.009) |

pretraining on ImageNet versus OpenImages, even when using more data. This result is in line with the observations from the main paper.

# D Class Activation Maps

Figure S2 shows class activation maps [32] obtained from training linear classifiers on top of frozen representations for ImageNet. The representations were obtained by training MoCo for 200 epochs on MS-COCO. The default two-crop (top rows) or multi-crop (bottom rows) transformations were used to generate positive pairs. We observe that the multi-crop model is better at localizing the object of interest. In particular, the two-crop model often attends to only a few parts of the object of interest. Differently, the class activation maps obtained with the multi-crop model seem to segment the complete object. We believe these results could be useful for researchers working on weakly-supervised semantic segmentation methods as well. The latter group of works used class activation maps to obtain a segmentation from annotations that are easy to obtain (e.g. image-level tags [20, 26]).

# E Video Instance Segmentation

Figure S3 shows additional qualitative results for the video instance segmentation task on DAVIS-2017. The input consists of an annotated frame. The annotations are propagated across frames by using nearest neighbors following [14]. Note that the evaluation does not require any finetuning. We conclude that the representations can be used to model dense correspondences in videos.

# F Semantic Segment Retrieval

Figure S4 shows additional qualitative results for the semantic segment retrieval task on PASCAL VOC (see also A.7). The input consist of a query region obtained from applying K-Means to the input image. We compute a region descriptor by averaging the features of all the pixels within the region. Finally, we retrieve the nearest neighbors for the query.

# G Limitations

We performed extensive experiments to measure the influence of different dataset biases on the representations. The experimental setup covered the use of object-centric vs. scene-centric data, uniform vs. long-tailed class distributions and general vs. domain-specific data. Further, we explored various ways of imposing additional invariances to improve the representations. Undoubtedly, there are several components that fall outside the scope of our study. We briefly discuss some of the limitations below.

**Other SSL methods.** In this paper, we performed our experiments using the MoCo framework. It still remains an open question whether our findings will also translate to other self-supervised methods like SimCLR [4], SWAV [2], BYOL [12], etc.

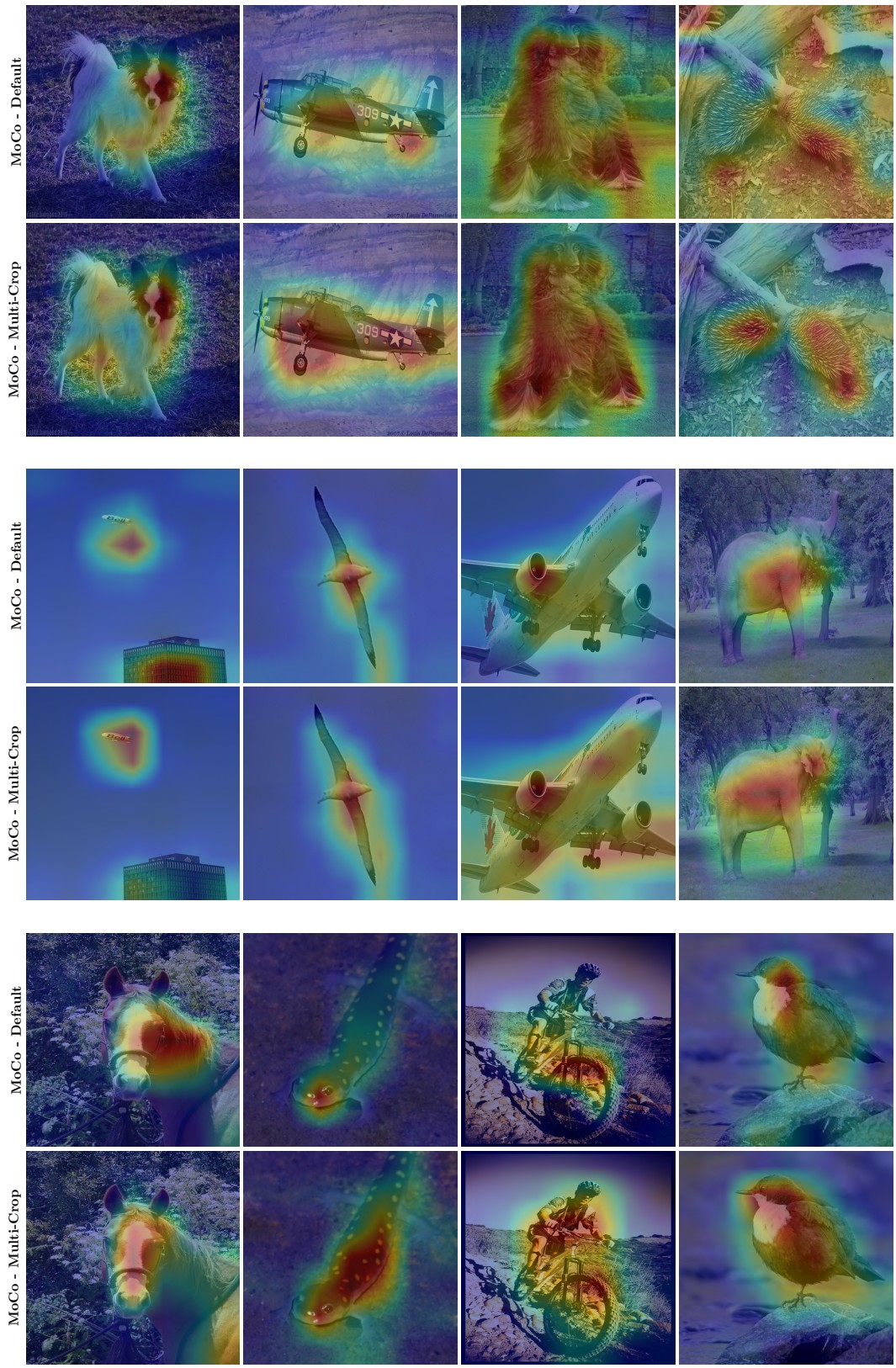

**Figure S2:** We show class activation maps for linear classifiers trained on top of frozen representations for ImageNet. Results are shown for MoCo trained with the two-crop (*top rows*) or multi-crop (*bottom rows*) transform. We trained for 200 epochs on MS-COCO.

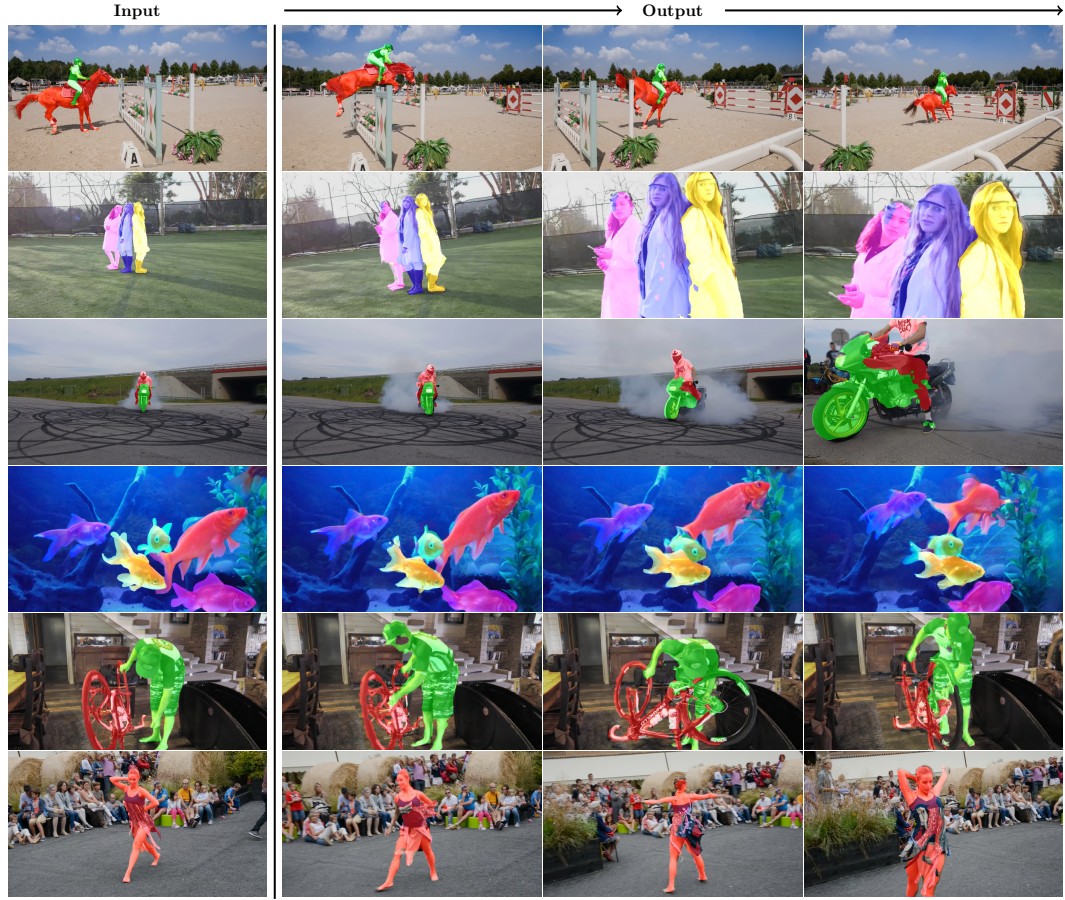

**Figure S3: DAVIS 2017 video instance segmentation.** Results are shown when training MoCo with the `multi-crop` transform for 200 epochs on MS-COCO. The labels are propagated for frozen representations via nearest neighbors.

**Dataset size.** The experiments in the main paper were conducted on datasets of moderate size (e.g. MS-COCO contains 118K images). It remains an open question whether the same results will be observed for larger datasets. We partially addressed this by repeating a subset of our experiments on datasets of increasing size in Section C. However, these experiments still only considered datasets with fewer than a million images. Moreover, it becomes increasingly difficult to isolate specific properties of the data when using more samples. For example, its not obvious that a uniform dataset of 1 billion images exists. In conclusion, we believe it would be useful to further investigate the behavior of existing methods on large-scale datasets.

**Data augmentations.** Imposing invariances to different data transformations proves crucial to learn useful representations. In this work, we used the same data augmentation strategy for all our experiments. However, different types of data could benefit from a different set of augmentations. We hypothesize that this is particularly true for domain-specific datasets. In this case, we expect that specialized data transformations, based upon domain-knowledge, could further boost the results. Further, we observe that the performance of existing methods still strongly depends on handcrafted augmentations, which can limit their applicability. We tried to partially alleviate this problem by adopting nearest neighbors - which only relies on the data manifold itself. Still, this problem could benefit from further investigation.

**Inductive biases.** All experiments were performed with a ResNet architecture. It would be interesting to study how architectural design choices influence the representation quality. In particular, one could study more general models like transformers [9, 28] which incorporate different biases.

**Query Region**

**Retrievals**

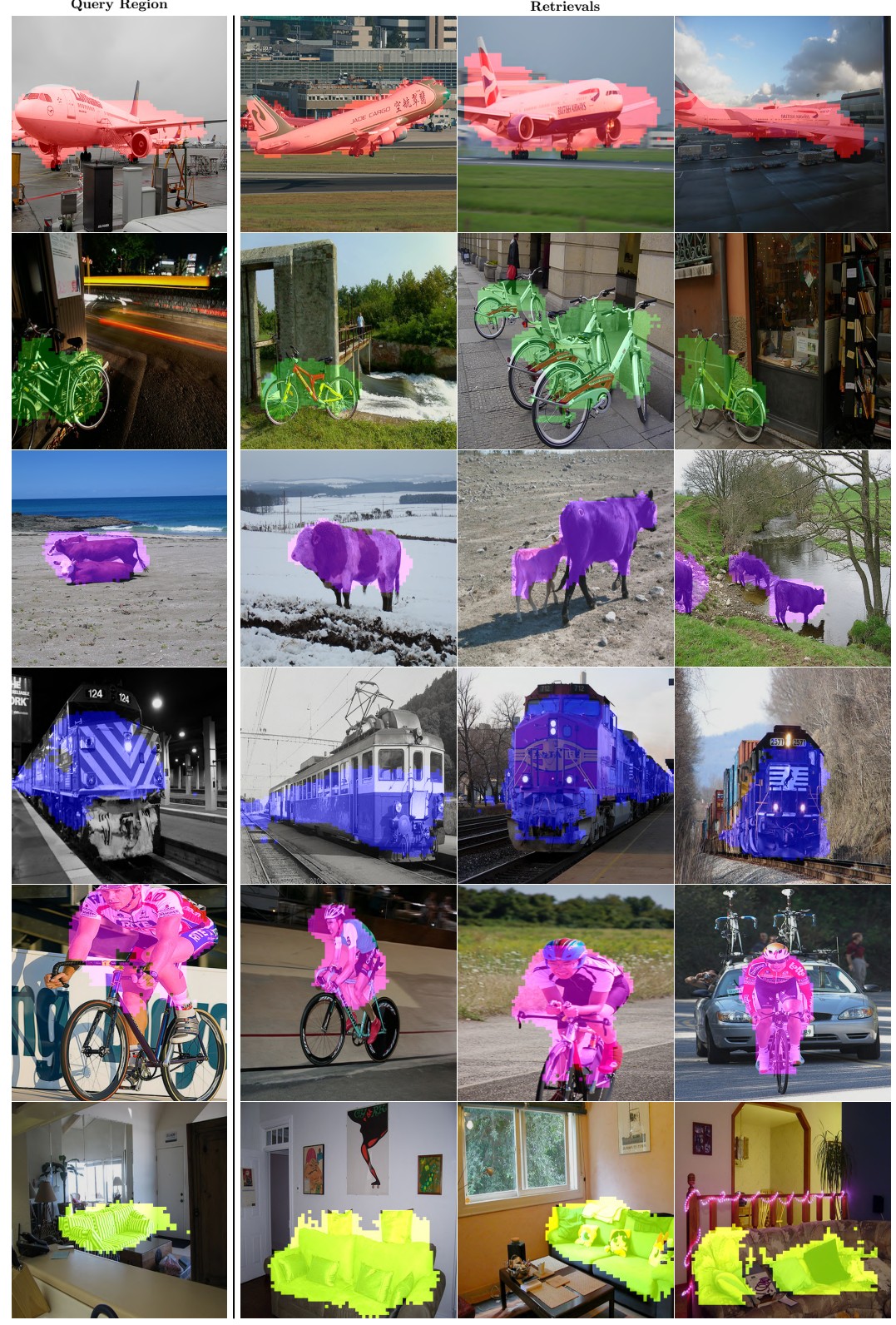

**Figure S4: PASCAL VOC semantic segment retrieval.** Results are shown when training MoCo with the `multi-crop` transform for 200 epochs on MS-COCO.