# OpenReview forum: "Revisiting Contrastive Methods for Unsupervised Learning of Visual Representations"
_NeurIPS.cc/2021/Conference — NeurIPS 2021 Poster_

### Official Review · Reviewer_syro · 2021-07-14

**Rating:** 7
**Confidence:** 4

**Summary:**

- Current contrastive methods mainly focus on curated iconic datasets like ImageNet.
- The paper studies the effect of dataset biases (like centered object, uniform data distribution) on performance of existing self-supervised learning methods.
- The paper shows that MoCo works well across a wide range of datasets. The paper analyses these datasets by separating them into different categories.
- In addition to the analysis, the paper proposes improvement by modifications to multi-scale cropping, stronger augmentations and a NN based methods. In particular, the authors observe that random cropping for instance discrimination tasks works well across scenarios.
- The paper shows that multi-crop transform leads to significant gains and gives rise to spatially structured representations which are useful for tasks like semantic segment retrieval and video instance segmentation.


**Limitations And Societal Impact:**

- The authors have adequately addressed the potential negative societal impact of their work.
- The supplementary material includes a discussion on the drawbacks of the proposed approach.


**Main Review:**

Strengths:
- The paper investigates the effect of dataset biases on self-supervised learning approaches. Further, the focus on uncurated datasets is an important, and use of these datasets for self-supervised learning has only started recently.
- The authors quantitatively verified that small IoU in scene centric dataset rarely occurs. As pointed out by the authors, this would effectively ensure that for decently sized objects, the two crops would have a significant overlap between the positives.
- The observation that transfer performance doesn't change much wrt changing Max IoU is also quite interesting.
- MC and CC forces the network to focus less on the discriminative parts but rather the whole object. This is also qualitatively shown using gradcam masks.
- The paper proposes a simple neat trick to mine for more examples by finding the nearest neighbors for anchors using features before the projection layer.
- The paper seems good improvements using a combination of the three proposed modifications to MoCo.
- The paper is an empirical one that shows some interesting analysis before delving into some easy fixes to improve performance.

Concerns:
- Would you have a significant memory penalty since you are storing pre projection features for the entire queue and these are typically of a larger dimension (2048D?) than the features after the final projection layer (typically 128D)
- The novelty of the proposed approaches is a bit lacking since it builds on top of methods that have been proposed previously (Multi-crop, Autoaugment). But, the combination of these approaches and the simple fixes to improve performance will be beneficial to the community. Further, the analysis on dataset biases and self-supervised learning will be beneficial to the community.
- The paper should mention the motivation behind choosing a 118K sized subset for ImageNet and OpenImages dataset. It looks like this is motivated from the size of the COCO dataset but it should be clarified.
- The authors note that the improvements start to reduce when the size of the dataset increases. It will be beneficial to show a $\Delta$ from a baseline trained with the same amount of data.
- L136 : I am curious if you have some experiments to show that this is indeed what's happening. While I understand that camera parameters would remain the same, wouldn't that be an incorrect signal for the self-supervised learning task, ie a shortcut that we don't want the network to rely on ? Further, for color/shape, it makes sense if the two crops are a part of the same object but that cannot be ensured for datasets like OpenImages.
- L145 : How do you explain the better performance of the long tailed model in Tables 2/3 ?
- L246 : should be $g(x^+)$ ? Also, the equation is a bit confusing. It might help to have a call to the projection layer explicitly here.


**Time Spent Reviewing:**

4

---

> ### Author Response · Authors · 2021-08-09
> **First Response To Reviewer syro**
>
> We thank the reviewer for his/her thoughtful feedback. We are encouraged that the reviewer finds that the paper provides an interesting analysis and acknowledges that the method shows good improvements. We address the reviewer's comments below and will incorporate all feedback.
>
> __1. Memory penalty of pre-projection features?__ Indeed, we keep an additional queue of 65 K samples with features of dimension 2048. This means we need to store an additional ~ 0.5 GB (65 K samples * 2048 floats/sample * 4 bytes/float). The default hardware setup for MoCo (8 x 16 GB GPU’s) can easily support this extra memory requirement. Therefore, we believe the memory penalty incurred by saving the pre-projection features does not present a significant weakness.
>
> __2. Novelty:__ We appreciate that the reviewer acknowledges that the novelty of our framework lies not in a single specific design choice, but is given by their composition: the use of multi-scale constrained cropping, stronger augmentations and k-Nearest Neighbors. Indeed, as the reviewer points out, the analysis on both the dataset biases and the use of additional invariances can be beneficial to the community.
>
> __3. Why use a 118 K sized subset for ImageNet/OpenImages?__ Indeed, we ensure that all datasets are of similar size to facilitate an apples-to-apples comparison between different pretraining datasets. MS-COCO is our smallest dataset with 118 K images. This explains why we use a 118 K sized subset for ImageNet and OpenImages. The dataset construction process can be found in the supplementary materials (see L9-L13). We will make sure to add a note about the dataset sizes in the main paper to improve transparency.
>
> __4. Results when increasing the dataset size:__ We refer to Section C in the supplementary materials for a concrete experiment on the effect of the dataset size. Indeed, perhaps not surprisingly, we observe diminishing returns when increasing the size of the dataset. Further, we find that the conclusions from Section 3 remain valid when increasing the size of the pretraining dataset. If anything is still unclear, we would be happy to answer any further questions.
>
> __5. Why does aggressive cropping still work?__ Indeed, the experimental results in Section 3.1 show that aggressive cropping is not detrimental for the transfer performance. So how can we explain this behavior? Well, patches within the same image were observed at the same point in time and space. As a result, they will often share useful information, which means the patches are correlated even for scene-centric images. Indeed, it is true that the patches won’t always share the exact same colors or shapes, but these cues will still be correlated. In conclusion, the point we wanted to make is that aggressive cropping does not yield views that share no useful information at all. If this were the case, it would not have been possible to learn useful representations. This viewpoint differs from other works like [38,39].
>
> __6. How do you explain the better performance of the long tailed model in Tables 2 & 3?__
> We believe this is due to the specific long-tailed version of ImageNet we sampled for our experiments. When averaging the results across multiple long-tailed versions of ImageNet, we observe that the results are closer to the ones obtained with a uniform version of ImageNet. We will report averaged numbers for the long-tailed models in future versions of the paper.
>
> __7. L246 - should be g(x+)?__ Indeed, the equation should use g(x+). We thank the reviewer for pointing this out.

---

> > ### Comment · Reviewer_syro · 2021-08-31
> > **Thanks for the rebuttal**
> >
> > Thanks for your responses! The rebuttal adequately addresses my concerns.
> >
> > Regarding some of the other key issues that were raised :
> >
> > - Novelty : I do agree that individual parts of the proposed approach are not too novel but the combination is. Further, as also explained by the authors as response to Reviewer KpV4, I would consider the presented analysis on invariances and biases to be novel.
> >
> > - Take-home-message : Reading the other reviews, I think this it is a valid concern. That said, I think that the clarification posted by the authors is convincing and would incorporating the proposed changes would make the message clear.
> >
> > - Use with other SSL methods : I think that the paper builds upon a fairly strong (contrastive) SSL baseline and given that most of the modifications/analyses are on the augmentation level, it might be reasonable to expect that it would work. That said, the authors might want to include more experiments to back up the hypothesis for the next draft.
> >
> > Overall, I like the presented analysis and the approach. I think the paper will be useful to the community. Based on the reviews, author response and responses from other reviewers I will retain my rating (7).

---

> > > ### Author Response · Authors · 2021-08-31
> > > **Thanks**
> > >
> > > We thank the reviewer for the valuable feedback.

---

### Official Review · Reviewer_1MyT · 2021-07-15

**Rating:** 6
**Confidence:** 4

**Summary:**

The paper studies the current stream of instance discrimination-based self-supervised learning algorithm (MoCo) beyond the typical object-centric dataset, i.e., ImageNet. Experiments are conducted on scene-centric datasets, long-tail datasets, and general-domain datasets. Results show that MoCo works well on these datasets. Furthermore, the authors studied multi-crop and stronger data augmentation for MoCo and showed that these enhanced augmentation techniques improved performance.

**Limitations And Societal Impact:**

The authors didn't discuss the limitations of the study. I find no potential negative societal impact.

**Main Review:**

Strengths:
-	The paper is clearly rewritten and very easy to follow.
-	The paper studies some of the essential problems in SSL. SSL models are trained mostly on ImageNet dataset, which by itself has inductive bias from data collecting procedure. Studying on other types of datasets, i.e., scene-centric and general-domain datasets, is an important steppingstone to better scale up the training of SSL models.
-	Experiments are thorough and convincing. Benchmarks are properly set up and follow the common practice.

Weaknesses:
-	Missing supervised baselines. Since most experiments are done on datasets of scale ~100k images, it is reasonable to assume that full annotation is available for a dataset at this scale in practice. Even if it isn’t, it’s an informative baseline to show where these self-supervised methods are at comparing to a fully supervised pre-trained network.
-	The discussion in section 3 is interesting and insightful. The authors compared training datasets such as object-centric versus scene-centric ones, and observed different properties that the model exhibited. One natural question is then what would happen if a model is trained on \emph{combined} datasets. Can the SSL model make use of different kinds of data?
-	The authors compared two-crop and multi-crop augmentation in section 4, and observed that multi-crop augmentation yielded better performance. One important missing factor is the (possible) computation overhead of multi-crop strategies. My estimation is that it would increase the computation complexity (i.e., slowing the speed) of training. Therefore, one could argue that if we could train the two-crop baseline for a longer period of time it would yield better performance as well. To make the comparison fair, the computation overhead must be discussed.
It can also be seen from Figure 7, for the KNN-MoCo, that the extra positive samples are fed into the network \emph{that takes the back-propagated gradients}. It will drastically increase training complexity as the network not only performs forward passing, but also the backward passing as well.
-	Section 4.2 experiments with AutoAugment as a stronger augmentation strategy. One possible trap is that AutoAugment’s policy is obtained by supervise training on ImageNet. Information leaking is likely.

Questions
-	In L114 the authors concluded that for linear classification the pretraining dataset should match the target dataset in terms of being object or-scene centric. If this is true, is it a setback for SSL algorithms that strive to learn more generic representations? Then it goes back again to whether by combining two datasets SSL model can learn better representations.
-	In L157 the authors discussed that for transfer learning potentially only low- and mid-level visual features are useful. My intuition is that low- and mid-level features are rather easy to learn. Then how does it explain the model’s transferability increasing when we scale up pre-training datasets? Or the recent success of CLIPs? Is it possible that \emph{only} MoCo learns low- and mid-level features?

Minor things that don’t play any role in my ratings.
-	“i.e.” -> “i.e.,”, “e.g.” -> “e.g.,”
-	In Eq.1, it’s better to write L_{contrastive}(x) = instead of L_{contrastive}. Also, should the equation be normalized by the number of positives?
-	L241 setup paragraph is overly complicated for an easy-to-explain procedure. L245/246, the use of x+ and x is very confusing.
-	It’s better to explain that “nearest neighbor mining” in the intro is to mine nearest neighbor in a moving embedding space in the same dataset.

Overall, I like the objective of the paper a lot and I think the paper is trying to answer some important questions in SSL. But I have some reservation to confidently recommend acceptance due to the concerns as written in the “weakness” section, because this is an analysis paper and analysis needs to be rigorous. I’ll be more than happy to increase the score if those concerns are properly addressed in the feedback.


**Time Spent Reviewing:**

2

---

> ### Author Response · Authors · 2021-08-09
> **First Response To Reviewer 1MyT**
>
> We thank the reviewer for his/her thoughtful feedback. We are encouraged that the reviewer finds the paper clearly written and easy to follow, the studied problem of high importance and the experiments convincing. We address the reviewer's comments below and will incorporate all feedback.
>
> __1. Comparison with supervised baseline:__ We perform supervised pretraining on the IN-118K dataset in order to compare the self-supervised methods with a supervised baseline. We will update the tables from the paper with the results below.
>
> **Table 2:** Comparison of linear classification models trained on top of frozen features.
>
> | Pretraining               | CIFAR10 | Cars | Food | Pets | SUN | VOC |
> |----------------------------|--------------|-------|---------|--------|--------|--------|
> | Supervised (IN-118K) | 83.1         | 36.2 | 54.5 | 87.5 | 47.2 | 79.3 |
> | MoCo (IN-118K) | 83.1 | 35.9 | 62.2 | 68.9 | 50.0 | 75.8 |
> | MoCo (COCO) | 77.4 | 33.9 | 62.0 | 62.6 | 53.6 | 80.9 |
>
> **Table 3:** Comparison of representations under the transfer learning setup.
>
> | Initialization | VOC Seg. | CS Seg. | NYUD Seg. | DAVIS | NYUD Depth |
> |-----------------|---------------|-------------|-----------------|-----------|-------------------|
> | Supervised (IN-118K) | 68.7 | 69.7 | 37.4 | 59.0 | 0.615 |
> | MoCo (IN-118K)         | 68.9 | 70.1 | 37.7 | 63.5 | 0.625 |
> | MoCo (COCO)           | 69.1 | 70.3 | 39.3 | 65.1 | 0.612 |
> | MoCo+ (COCO)             | **73.5** | **72.3** | **41.3** | **66.2** | **0.580** |
>
> We compare the results of the supervised pretraining (Supervised IN-118K) with MoCo (MoCo IN-118K and MoCo COCO). MoCo obtains better results under the transfer learning setup (see Table 3). The representations are further improved when training MoCo with additional invariances (see MoCo+). For linear classification, we observe a more mixed view. The supervised baseline results in the best performance when the classes in the target dataset align with the ImageNet classes. For example, a large number of the classes from the Pets dataset are also present in ImageNet, which could explain the performance gap (Supervised: 87.5 versus MoCo: 68.9). The situation is reversed when the target classes show less overlap with the ImageNet classes. For example, the Food dataset features more fine-grained classes than the ones present in ImageNet (Supervised: 54.5 versus MoCo: 62.0).
>
> __2. Can we combine different sources of data?__ Yes, a recent (unpublished) work [A] has studied this particular setup. The authors explore a hierarchical pretraining scheme, which sequentially pretrains on a large, general, base dataset like ImageNet and then on specialized datasets that are similar to the target data. We argue that the conclusions from [A] complement our analysis from Section 3. In particular, we focus on the influence of different datasets, while [A] studies how to optimally combine different datasets. We will add a reference to this work for completeness.
>
> [A] Reed, Colorado J., et al. "Self-Supervised Pretraining Improves Self-Supervised Pretraining." arXiv preprint arXiv:2103.12718. 2021.
>
> __3. Two-crop versus multi-crop:__ This question can be answered from the results in Figure 5. Let us reiterate. Figure 5 plots the training time versus performance for the two-crop and multi-crop augmentation. We evaluate the representations for the classification task on PASCAL VOC. We draw the following two conclusions.
>
> First, the multi-crop yields better performance than the two-crop augmentation for a specific training budget. Thus, the multi-crop improves the computational efficiency w.r.t. the two-crop transfrom [see L190-L191]. Second, even when increasing the training duration for the two-crop augmentation, e.g., from 200 to 800 epochs, we obtain better results with the multi-crop augmentation. This shows that the overall improvements obtained with the multi-crop augmentation can not be attributed to increasing the training budget.
>
> __4. AutoAugment:__ Indeed, AutoAugment is an augmentation policy obtained through supervised learning on ImageNet, as we point out in the paper [see L218-L220]. However, we believe this does not impose any significant limitations for our approach because of the following.
>
> First, we show that the augmentation strategy can be applied to a scene-centric dataset, e.g., MS-COCO, even though the policy was obtained through supervised learning on ImageNet. This means that the policy transfers across datasets - which guarantees its applicability for self-supervised learning. Second, we show that our gains translate to multiple downstream tasks and not only to the ImageNet classification task for which AutoAugment was optimized. Third, other alternatives like RandAugment [B] could be used as a drop-in replacement for AutoAugment as shown in [C].
>
> [B] Cubuk, Ekin D., et al. "Randaugment: Practical automated data augmentation with a reduced search space." CVPR. 2020. \
> [C] Sohn, Kihyuk, et al. "FixMatch: Simplifying Semi-Supervised Learning with Consistency and Confidence." NeurIPS. 2020.
>
> __5. Question - Linear classification:__ In linear classification, the feature representations are fixed. Therefore, it is beneficial to ensure that the source and target distributions are aligned. The results in Table 2 confirm this; e.g., pretraining on an object-centric dataset results in better performance when the target dataset is object-centric. Differently, in Table 3, we observe that it’s less important to align the source and target distributions for end-to-end finetuning. We refer to our first point for a discussion on how to combine multiple datasets.
>
> __6. Question - Low-and mid-level visual features:__ Let us first discuss what we mean by low- mid- and high-level features and what features are important for linear classification/transfer learning. The high-level features pertain to the semantics of a particular dataset. The alignment between the high-level features of the source and target dataset are important under the linear classification setting, where we observe that better alignment results in better performance (see Table 2). However, when it comes to end-to-end finetuning, we observe similar results even when the mismatch between source and target dataset increases (see Table 3). The latter suggests that mostly the low-and mid-level features are kept, while the high-level features quickly adapt to the target dataset.
> Based upon the above discussion, we hypothesize that MoCo is able to learn useful low-, mid- and high-level features. At the same time, we observe that the low- and mid-level features matter more for transfer learning, while the high-level features matter more for linear classification. Again, this is supported by the empirical evidence in Tables 2-3.
>
> Indeed, the model’s transferability increases when using a larger dataset (see Section C of the supplementary materials). However, we observe diminishing returns when increasing the size of the dataset. As the size of the dataset increases, the model needs to capture more of the instance specific details of the image to solve the instance discrimination task. We hypothesize that this forces the model to capture additional mid-level information, which results in improved transferability.
>
> Regarding the comparison with CLIP, we believe it is hard to draw any fair comparisons between our work and CLIP. In particular, the training setup in CLIP is very different as it uses image-caption pairs, a different architecture, a larger dataset, etc.
>
> __7. Limitations And Societal Impact:__ The limitations of our study are discussed in Section G of the supplementary materials.

---

> > ### Comment · Reviewer_1MyT · 2021-08-27
> > **Response**
> >
> > The authors have adequately addressed some of my concerns, despite not all, in their response to my initial review. I also agree with another of my fellow reviewer that the message of the original draft can be improved. Regardless, I'm willing to increase the rating to 6.

---

> > > ### Author Response · Authors · 2021-08-27
> > > **Thanks**
> > >
> > > We thank the reviewer for increasing his/her rating. We will incorporate all feedback and make specific adjustments to sharpen the message of the paper (see response to R1).

---

### Official Review · Reviewer_vQPG · 2021-07-17

**Rating:** 7
**Confidence:** 3

**Summary:**

This paper presents rigorous analyses of self-supervised learning from different aspects. Here are their findings:

1. The pretraining dataset should match the target dataset in terms of being object- or scene-centric.
1. MoCo is robust to changes in the class distribution of the dataset (pre-trained on uniform ImageNet vs long-tailed ImageNet)
1. The model trained on data with discriminative features (ImageNet) performs equally well with the model trained on data without (BDD100K) in the fine-tuning setting for semantic segmentation tasks.

The authors also propose three strategies to improve MoCo:
1. Multi-scale constrained cropping data augmentation
1. Mixture of standard augmentation and AutoAugment
1. Including nearest neighbors as positive pairs

**Limitations And Societal Impact:**

- (copy and paste from above) Is the proposed strategies in Sec. 4 generalizable to other SSL methods? For example, a clustering-based method like SwAV.

- It is still unclear what kind of feature, bias, invariance, etc, the model captures when pre-trained on different datasets given only the performance in the linear evaluation setting and the fine-tuning setting presented in Sec. 3. For example, does the model learn a more robust representation? Is the model biased to shape or to texture?

**Main Review:**

### Strength
- This paper provides comprehensive and rigorous studies for different aspects of the pre-training data being used for SSL.
- Results in Tab. 2 challenge the observation in [38, 39]. The authors identify that the non-overlapping between views is in fact not the major issue. What really matters is to match the pre-training and the target dataset in terms of being object- or scene-centric. Analyses in Fig. 1 and Fig. 2 further support their claims.
- The proposed augmentation and training strategies in Sec. 4 are effective for improving MoCo's performance.

### Weakness

- The connection between the first part of the paper (Sec. 3) and the second part of the paper (Sec. 4) is weak. It looks like a concatenation of two papers. The reviewer expects to see how the analyses from Sec. 3 motivate the strategies proposed in Sec. 4.

- Is the proposed strategies in Sec. 4 generalizable to other SSL methods? For example, a clustering-based method like SwAV.

- For fine-tuning results, the reviewer suggests including baseline methods of random init for reference.

- In Sec. 3.3, the experiment of BDD on linear evaluation is missing. Is it possible that training on BDD has a negative effect in the linear evaluation setting?

- In Sec 4.2, the authors only explain why using AutoAugment leads to worse results. The explanations/analyses of why standard + auto-augment leads to better performance are not provided.

**Time Spent Reviewing:**

4

---

> ### Author Response · Authors · 2021-08-09
> **First Response To Reviewer vQPG**
>
> We thank the reviewer for his/her thoughtful feedback. We are encouraged that the reviewer finds that our work provides a comprehensive and rigorous study for various aspects of the pretraining data, challenges observations from prior works and proposes effective strategies for improving MoCo. We address the reviewer's comments below and will incorporate all feedback.
>
> __1. Connection Sec. 3 & Sec. 4:__ Based upon the results from Section 3, we try to improve MoCo by imposing additional invariances in Section 4. The motivation for this strategy is as follows.
>
> In Section 3, we show that MoCo works well for a large variety of pretraining datasets: scene-centric, long-tailed and domain-specific images. Based upon this result, we decide that there is no obvious need to develop a more advanced pretext task [39] or use an alternative source of data like video [38] to improve the representations. Instead, it makes sense to stick with the approach from MoCo - as it already proves to work well - and try to devise simple ways to boost the performance. So how can we realize this objective? The overall idea in recent SSL works is to learn representations by imposing invariances between positive pairs. The importance of mining useful positive pairs is also reflected in our analysis in Section 3. Therefore, in Section 4, we follow the strategy of finding better positive pairs to improve MoCo. To this end, we explore three different mechanisms: multi-scale crops, stronger augmentations and nearest neighbors. In each case, we measure the improvements w.r.t. the MoCo baseline and try to understand how the added invariances alter certain properties of the representations.
>
> We will make sure to add this motivation to the paper, so our strategy is spelled out more clearly.
>
> __2. Generalization to other SSL methods:__ Current state-of-the-art contrastive SSL methods like SimCLR [5], SwAV [4], MoCo [19], SimSiam [9], BYOL [17], etc., follow a rather similar setup. In particular, these methods share the same underlying structure, but adopt different design choices, as shown by [9]. For example, MoCo uses a memory bank, while SimCLR relies on a large batch size to keep a sufficient number of negatives. Given the similarities between these works, we expect that our improvements will also translate to other contrastive learning pipelines. In fact, we already observed similar improvements when adding the invariances from Section 4 to SimCLR [5] or DenseCL [45]. Further, we stress that MoCo provides a strong baseline, as it outperforms many competing works across multiple downstream tasks (see [9]).
>
> __3. Results with random initialization:__ The results with random initialization are provided below. The pretraining clearly has a positive effect on the performance. Note that the numbers with random initialization were already provided in Table 8. We will make sure to include them in Table 3 for completeness, as suggested by the reviewer.
>
> | Initialization | VOC Seg. | CS Seg. | NYUD Seg. | VOC Det. | DAVIS | NYUD Depth |
> |----------------|-----------------|-------------|-----------------|---------------|----------|-------------------|
> | Rand Init. | 39.2 | 65.0 | 24.4 | 38.0 | 40.8 | 0.867 |
> | MoCo IN-118K | **68.9** | **70.1** | **37.7** | **53.0** | **63.5** | **0.625** |
>
> __4. BDD on linear evaluation:__ BDD100K is a domain-specific dataset. Differently, MS-COCO, ImageNet and OpenImages are domain-agnostic datasets. Based upon this grouping, we decided to evaluate the representations from BDD100K only for a similar domain (i.e. Cityscapes). After all, a practioner using domain-specific data will be mostly interested in transferring the representations to a similar domain. For example, it's unlikely that a person would pretrain on a car dataset if he/she is interested in recognizing different types of food.
>
> To answer the reviewer's question, we provide the results below. The model pretrained on BDD100K reports lower performance on all datasets. This is no surprise. The classes of the target datasets are not present in a domain-specific dataset like BDD100K. As a result, the model does not learn the high-level features that are required to solve the linear classification tasks. Differently, MS-COCO and IN-118K feature more diverse classes from which the linear classification models benefit. We will add this discussion to the paper.
>
> | Pretraining | CIFAR10 | Cars | Food | Pets | SUN | VOC |
> |----------------|--------------|-------|---------|--------|--------|--------|
> | BDD | 71.8    | 18.0 | 40.1 | 35.5 | 37.8 | 54.5 |
> | IN-118K| 83.1 | 35.9 | 62.2 | 68.9 | 50.0 | 75.8 |
> | COCO | 77.4 | 33.9 | 62.0 | 62.6 | 53.6 | 80.9 |
>
>
> __5. Improvements with standard + auto-augment:__ We use AutoAugment to augment half of the samples in the batch. The data transformations in AutoAugment differ from the ones in the standard augmentation policy, which enforces the model to learn additional invariances. Therefore, the improved performance can be attributed to the use of a richer set of augmentations.
>
> __6. Limitations:__ We thank the reviewer for the suggestions. We will incorporate these notes in the limitations section of our work (Section G). To analyze the dependence of the visual representations on shape or texture, we could conduct a similar set of experiments as in [A]. For example, we could measure the image classification performance on a stylized version of ImageNet to study the dependence on texture.
>
> [A] Geirhos, Robert et al., ImageNet-trained CNNs are biased towards texture; increasing shape bias improves accuracy and robustness, ICLR, 2019.

---

### Official Review · Reviewer_KpV4 · 2021-07-19

**Rating:** 6
**Confidence:** 4

**Summary:**

The paper provides a comprehensive study and a set of improvements for the self-supervised method MoCo. First, the authors challenge the common assumptions that the training dataset has to be object-centric, not long-tailed and the crops overlapping, which provides very useful insights. Then, the authors propose a bag of tricks to improve the performance of MoCo in practice.

**Ethics Review Area:**

["I don’t know"]

**Limitations And Societal Impact:**

-

**Main Review:**

### Good:

- Very interesting and practically useful analysis of MoCo in Section 3. I would love to see a more extensive study on this matter in the paper.
- Extremely useful for practitioners bag of tricks presented in Section 4. I particularly appreciate k-NN positives presented in Section 4.3.

### Bad:

- Submission to the wrong venue. While the paper contains very useful content, there is no technical novelty at all, rather a set of experiments applying one model to different datasets and providing a bag of tricks for better performance. This would be a perfect fit for representation learning workshop, or perhaps a purely vision conference.
- The paper is lacking a holistic message. While Section 3 studies the inner working of MoCo and how it performs in different settings, Section 4 proposes technical improvements that have little to do with discoveries in Section 3. The paper has a feel of "we've made these experiments and wrote about all of them in this paper".
- It is not clear whether the improvements will translate to other self-supervised models rather than MoCo and whether it will translate to longer training schedules on full-size training datasets, i.e. the mode where SoTA self-supervised results are reported.

### Questions:

- It is surprising for me that having littler overlap between crops in scene-centric training datasets does not affect the performance on the target dataset (see Figure 2-3). While I can understand this effect to some extent when testing on a scene-centric dataset, what would happen if the test dataset is object-centric?

**Time Spent Reviewing:**

3

---

> ### Author Response · Authors · 2021-08-09
> **First Response To Reviewer KpV4**
>
> We thank the reviewer for his/her feedback. We are encouraged that the reviewer finds our paper very interesting and extremely useful. We address the reviewer's comments below and will incorporate all feedback.
>
> __1. Submission to the wrong venue:__
> We respectfully disagree and point out that all reviewers were positive about the paper. In particular, they find the paper interesting (R4), important (R3, R4), extremely useful (R1) and effective (R2). We firmly believe that our findings and analysis on the topic of self-supervised representation learning are of relevance to the conference’s audience. Further, we disagree with the statement that this paper does not contain any (technical) novelty at all. We argue that our analysis on the influence of dataset biases and the use of additional invariances is novel. Our paper takes a more experimental point of view, as do many other papers on self-supervised learning at machine learning conferences (e.g., [15, 38, 54, A]).
>
> We list our most important contributions below to address the reviewer’s concerns regarding novelty of the paper:
>
> - We study the influence of different dataset biases on the learned representations in Section 3. In contrast to prior work [38,39], our experimental results show that MoCo works well for both object- and scene-centric datasets. Further, we dispute several claims from recent works [38,39] regarding the use of non-overlapping crops or the need for additional object priors in SSL (see Section 3.1). We point out that all reviewers were positive about this analysis.
>
> - We find several effective ways to improve the learned representations (see Section 4). These include multi-scale constrained cropping (Section 4.1), stronger augmentations (Section 4.2) and the use of a kNN loss (Section 4.3). Both the combination and analysis of these components is novel. Also, the proposed implementation of the kNN loss is novel, which was also appreciated by the reviewer.
>
> - Finally, we uncover several interesting properties of the learned representations (see section 4.1 - Table 5 & 6). In particular, when training the model with multi-scale constrained crops, we find that the representations can be directly used for video object segmentation and semantic segment retrieval. We believe this analysis is novel and of relevance to the community.
>
> We would be happy to address any other concerns regarding novelty if the reviewer provides us with more specific points or works.
>
> [A] Zoph, Barret, et al. "Rethinking Pre-training and Self-training." NeurIPS (2020).
>
> __2. Holistic message:__ We apologize if the motivation for our strategy was not sufficiently clear. We spell out our motivation in more detail below.
>
> In Section 3, we show that MoCo works well for a large variety of pretraining datasets: scene-centric, long-tailed and domain-specific images. Based upon this result, we decide that there is no obvious need to develop a more advanced pretext task [39] or use an alternative source of data like video [38] to improve the representations. Instead, it makes sense to stick with the approach from MoCo - as it already proves to work well - and try to devise simple ways to boost the performance. So how can we realize this objective? The overall idea in recent SSL works is to learn representations by imposing invariances between positive pairs. The importance of mining useful positive pairs is also reflected in our analysis in Section 3. Therefore, in Section 4, we follow the strategy of finding better positive pairs to improve MoCo. To this end, we explore three different mechanisms: multi-scale crops, stronger augmentations and nearest neighbors. In each case, we measure the improvements w.r.t. the MoCo baseline and try to understand how the added invariances alter certain properties of the representations.
>
> In summary, the overall conclusions of this paper are two-fold. First, we show that MoCo obtains robust results across different datasets. Second, we improve the representations by imposing additional invariances. Both insights are useful for researchers working on self-supervised learning. In particular, both points illustrate that a simple pretext task like MoCo allows to learn state-of-the-art visual representations across multiple datasets. This strategy diverges from recent works [38,39,45] that proposed the use of more advanced pretext tasks.
>
>
> __3. Generalization to other SSL methods:__ Current state-of-the-art contrastive SSL methods like SimCLR [5], SwAV [4], MoCo [19], SimSiam [9], BYOL [17], etc., follow a rather similar setup. In particular, these methods share the same underlying structure, but adopt different design choices, as shown by [9]. For example, MoCo uses a memory bank, while SimCLR relies on a large batch size to keep a sufficient number of negatives. Given the similarities between these works, we expect that our improvements will also translate to other contrastive learning pipelines. In fact, we already observed similar improvements when adding the invariances from Section 4 to SimCLR [5] or DenseCL [45]. Further, we stress that MoCo provides a strong baseline, as it outperforms many competing works across multiple downstream tasks (see [9]).
>
> __4. Longer training schedules:__ Our results are expected to translate to longer training schedules on full-size training datasets. Table 9 already shows that we obtain similar improvements when increasing the training time (we increase the number of epochs from 200 to 800). Further, we point out that competing works like CAST [39], VirTex [13] and DenseCL [45] also pretrain on MS-COCO for 800 epochs.  We conclude that our experimental setup is no different.
>
>
> __5. Question - Evaluation on object-centric datasets:__ We perform an additional experiment to answer the reviewer’s question. In particular, we train a linear classifier on top of frozen representations, when limiting the overlap between crops to 0.5 during pretraining. Three different target datasets are use. These include CIFAR10, Pets and StanfordCars. The results are shown below. We conclude that there are no significant differences in the transfer performance.
>
> Max Overlap |0.5 | 1.0 |
> ------------------|-----|------|
> Pets               | 62.0 | 62.4 |
> CIFAR10       | 82.7 | 83.1 |
> Cars          | 33.3 | 33.9 |

---

> > ### Comment · Reviewer_KpV4 · 2021-08-11
> > **The message has to be sharpened**
> >
> > I thank the reviewers for providing a detailed and informative response.
> >
> > I agree with the authors and other reviewers that the content of the paper is valuable to the community and I agree that there has been a lot of empirical work published at NeurIPS before. Thus, I am ready to reconsider my decision about the paper.
> >
> > However, the global message of the paper is lacking clarity (which was also pointed out by Reviewer vQPG), i.e. The two main sections (Section 3 and 4) seem almost independent from each other. Even if the "overall conclusion of the paper is twofold", both conclusions must be connected by a global goal.  The authors must put more work into sharpening the message and connecting those parts. This will make the paper look less like a bag of tricks and more like a useful guide to self-supervised learning.
> >
> > I am willing to raise my rating of the paper if the authors provide a satisfactory strategy for mitigating the aforementioned problem.
> >
> > P.S. The response provided by the authors in point 2 is a step in the right direction but still does not seem convincing.

---

> > > ### Author Response · Authors · 2021-08-11
> > > **Sharpening of the message**
> > >
> > > We thank the reviewer for his/her response. We appreciate that the reviewer is willing to revise his/her opinion. To address the reviewer’s concern w.r.t. the take-home message of the paper, we start from the overall question this paper is trying to answer.
> > >
> > >
> > > __Research question__: *‘How can we learn more effective representations through contrastive self-supervised learning without relying too much on specific dataset biases?’*
> > >
> > > *The main take-away is that we can find a generic set of augmentations/invariances that allows us to learn effective representations across different types of datasets (i.e., scene-centric, non-uniform, domain-specific)*. We provide empirical evidence to support this claim. First, in Section 3, we show that the standard SimCLR augmentations can be applied across several datasets. Then, Section 4 studies the use of additional invariances to improve the results for a generic dataset (i.e. MS-COCO). The results in this paper are important, as they reduce the need for dataset/domain-specific expertise to learn useful representations in a self-supervised way. Instead, the results show that simple contrastive frameworks apply to a wide range of datasets. We believe this is an encouraging result. Finally, our overall conclusion differs from a few recent works [38, 39], which investigated the use of a more advanced pretext task or video to learn representations in a generic way.
> > >
> > > Our take-away message also yields a few interesting follow-up questions:
> > > * *Modalities*: What about other modalities like video, text, audio, etc.? Can we reach a similar conclusion?
> > > * *Invariances*: What other invariances can be applied? Can we control certain properties of the representations by focusing on specific invariances? For example, how do we bias the representations to focus more on texture, shape, etc.?
> > > * *Compositionality*: Can we combine different datasets?
> > >
> > > Finally, we propose the following strategy to ensure that our paper better reflects the global question and conclusion of the paper.
> > > * __Abstract + Introduction__: We will include the overall research question of the paper more explicitly. We will add a motivation for our overall strategy to the final paragraph of the introduction.
> > > * __Section 4__: We will add a motivation for imposing additional invariances to improve the representations, based upon the results from Section 3. The motivation can be found in our previous answer.
> > > * __Conclusion__: We will add the main take-away message and possible follow-up questions (see above). In this way, the conclusion will answer our overall question and connect the conclusions from Section 3 and 4 through a holistic message.
> > >
> > > We hope our response addresses the reviewer’s concerns. Of course, we would be happy to discuss any remaining questions or specific points related to our strategy or the take-home message of the paper.

---

> > > > ### Comment · Reviewer_KpV4 · 2021-08-27
> > > > **Great**
> > > >
> > > > I thank the authors for the work put into sharpening the message. It does look much more compelling now! In my opinion, this can improve the paper a lot and help spread a simple and effective message on how to do better in one of the hottest topics in ML now.
> > > >
> > > > I recommend the paper for acceptance.

---

> > > > > ### Author Response · Authors · 2021-08-27
> > > > > **Thanks**
> > > > >
> > > > > We thank the reviewer for his/her suggestions. We will make the proposed changes as outlined above.

---

### Decision · Program_Chairs · 2021-09-27

**Decision:**

Accept (Poster)

**Comment:**

This paper had thorough reviews, rebuttals, and discussion. The reviewers all agreed that the work presents a very interesting analysis and set of conclusions, along with some simple methods to improve results of MoCo-learned representations. While individual contributions are simple and not necessarily novel from a technical perspective, as agreed by the reviewers the set of narratives, experiments, interesting findings, and methods globally provide an interesting novel perspective to the community. Analysis of invariances and especially dataset types (e.g. object/scene-specific) is indeed interesting and one that may exist as common knowledge but has not been thoroughly and rigorously analyzed. In other words, while the paper is largely an empirical investigation it is well-executed in making claims/hypothesis and resulting analysis.

One of the main concerns expressed by multiple reviewers is the lack of a strong connection and holistic perspective offered across the two sections of the paper. In the rebuttal, the authors provided several arguments and expanded this narrative and connections, which the reviewers were satisfied with. The authors should comprehensively incorporate this (and other great suggestions made by the reviewers) into the final version. While not mandatory, one of the remaining weaknesses that have not been addressed is the applicability of the findings to other self-supervised learning methods. We encourage the authors to add this if at all possible, as it would significantly increase the impact of the paper.